# EvoCF: Multi-agent Collaboration with Memory-guided Evolutionary Counterfactual Planning

## Abstract

Planning collaboration strategies for multi-agent embodied systems remains a core challenge for LLM-based planners, which often fail to capture the physical and coordination constraints of real-world environments. To address this, we present **EvoCF** (Evolutionary Counterfactual Planning), a memory-guided framework for discovering improved multi-agent collaboration strategies through counterfactual plan generation and evaluation. First, we induce a structured symbolic rule library from failure experiences, encoding reusable constraints of inter-agent dependencies and action feasibility. Then, we propose an evolutionary counterfactual plan generator that systematically explores semantically consistent plan variants through rule-guided mutations. This enables the discovery of robust multi-agent strategies beyond short-sighted LLM plans. Finally, we design an experience-driven evaluator that scores candidate plans along multiple metrics, using retrieval-augmented constraint matching. Across embodied simulation benchmarks, EvoCF consistently discovers more robust and executable plans compared to baseline approaches. Our results demonstrate that grounding multi-agent planning in structured memory and symbolic reasoning significantly enhances both reliability and adaptability.

## 1 Introduction

Enabling multiple embodied agents to collaborate effectively in complex environments is a central challenge for artificial intelligence, with far-reaching implications for robotics, household assistance, and industrial automation. Unlike single-agent planning, multi-agent collaboration requires agents not only to sequence their own actions but also to coordinate their roles, respect inter-agent dependencies, and adapt to the physical and communication constraints of real-world environments. Designing planners that can reliably discover and execute such joint strategies remains a core bottleneck in embodied AI.

Large language models (LLMs) have recently been applied as high-level planners due to their ability to interpret natural language instructions and generalize across tasks. Early efforts have extended LLM planners to multi-agent settings through various paradigms. For example, Kannan et al. (2024) introduce a prompting framework (SMART-LLM) that decomposes high-level missions into coalitions and task allocations via few-shot examples. Other work integrates LLM planning with iterative feedback loops: Nayak et al. (2024) propose an LLM-based plan–act–correct–verify loop (LLa-MAR) that uses execution feedback for self-correction. Similarly, Zhang et al. (2024a) develop a modular approach (CoELA) that combines LLM reasoning with perception, long-term memory, and communication, enabling agents to coordinate and assist each other in a shared environment. At a larger scale, Qian et al. (2025) organize swarms of agents into directed acyclic graphs (MacNet), revealing that performance follows a logistic growth law as team size increases. These advances demonstrate the promise of LLM-based planners for multi-agent tasks, but also highlight key limitations: current systems typically produce a single plan at a time without systematic revision, rely on heuristic corrections or centralized controllers, and rarely consider alternative task assignments or counterfactual re-planning.

These limitations are further amplified in dynamic, partially observable environments (POMDP). While in single-agent settings, incorporating feedback and re-planning, e.g., via few-shot prompting as in LLM-Planner (Song et al., 2023), improves robustness, extending such approaches to multi-agent scenarios under partial observability is non-trivial. The joint action space scales exponentially with the number of agents, and effective coordination requires reasoning over uncertain observations, asynchronous actions, and long-range temporal dependencies such as task coupling and load balancing. We argue that robust multi-agent collaboration requires explicit counterfactual reasoning at planning time. In other words, agents should continuously ask "what if?": What if a different agent were assigned to this subtask? What if the order of two actions were swapped? What if a prerequisite action is skipped? Counterfactual reasoning has been explored in multi-agent reinforcement learning for credit assignment (e.g., the COMA algorithm uses a counterfactual baseline to isolate an agent's contribution (Foerster et al., 2018)). In the context of LLM-based planners, however, systematic counterfactual exploration of plan alternatives is largely absent.

In this paper, we propose EvoCF, a memory-guided evolutionary planning framework that addresses a key gap in multi-agent embodied AI by combining structured memory, symbolic constraint induction, and counterfactual search. EvoCF grounds planning in long-term memory of past execution traces, organized by tasks, agents, and outcomes. During planning, it retrieves relevant experiences and induces symbolic constraints capturing physical feasibility and coordination requirements. These constraints form a rule library that guides plan generation (see Appendix B.3). EvoCF uses an evolutionary search procedure to iteratively refine joint plans by mutating the original plan, while enforcing semantic consistency with retrieved rules. A memory-guided evaluator scores candidate plans across criteria such as predicted success, physical validity, inter-agent synchronization, and efficiency. To evaluate EvoCF, we use MAP-THOR(Nayak et al.), a large-scale simulation benchmark for multi-agent embodied rearrangement tasks under partial observability. It features realistic apartment layouts, diverse object configurations, and multiple agents with asynchronous actions, posing significant challenges for coordination, spatial reasoning, and task planning. By exploring diverse counterfactual variants, EvoCF discovers collaboration strategies that are more robust and adaptable than the single-shot outputs of reactive LLM planners.

This work makes the following contributions:

- We introduce EvoCF, which grounds multi-agent planning in a structured memory of past experiences. By retrieving relevant traces and inducing symbolic rules from failures, EvoCF builds a library of reusable constraints that guides plan generation and evaluation.
- We design an evolutionary counterfactual plan generator that systematically explores alternative joint plans by mutating action assignments and orderings. This enables discovery of robust collaboration strategies beyond conventional one-shot LLM plans.
- We propose an experience-driven evaluator that ranks candidate plans using both learned constraints and retrieved outcomes. This evaluator leverages stored experiences to anticipate the consequences of actions and select effective multi-agent strategies.
- Through extensive experiments on multi-agent embodied simulation benchmarks, we demonstrate that EvoCF finds more reliable and executable plans than state-of-the-art baselines. EvoCF achieves 18% higher success rate over LLaMAR and significantly improves other metrics such as transport rate, coverage, balance, and planning efficiency.

## 2 RELATED WORK

**LLM-based multi-agent planning.** Large language models have recently been extended to multi-agent planning. SMART-LLM (Kannan et al., 2024) decomposes missions into subtasks and allocations via prompting. LLaMAR (Nayak et al., 2024) employs a plan–execute–monitor–repair cycle, boosting performance on MAP-THOR and Search&Rescue tasks. CoELA (Zhang et al., 2024a) integrates LLM reasoning with perception, memory, and communication for emergent cooperation, while MacNet (Qian et al., 2025) scales collaboration through DAG-based hierarchies, showing logistic gains with larger teams. Other paradigms differ in centralization: CoNavGPT (Yu et al., 2023) produces global plans from a single GPT model, whereas RoCo (Mandi et al., 2024) assigns each agent its own LLM controller with natural-language communication.

**Structured memory and symbolic knowledge.** Incorporating memory improves long-horizon planning. Generative Agents (Park et al., 2023) and Voyager (Wang et al., 2023) show how agents can recall or reuse past trajectories to enhance coherence and skill acquisition. More recently, struc-

tured memory systems such as MIRIX (Wang & Chen, 2025), Intrinsic Memory Agents (Yuen et al., 2025), and TME (Ye, 2025) introduce richer episodic and role-aligned components, improving robustness on long-horizon tasks. Reflexion (Shinn et al., 2023) further enables iterative self-critique and refinement. Within multi-agent contexts, CoELA (Zhang et al., 2024a) integrates episodic memory but still lacks symbolic abstraction.

**Counterfactual reasoning and plan optimization.** Recent work explores counterfactual reasoning in LLMs through diverse mechanisms for generating and evaluating alternative plans. Tree-of-Thoughts (Yao et al., 2023) explores multiple reasoning paths, while ReAct (Yao et al., 2022) interleaves reasoning and acting. Extensions include CFMAD, where agents debate opposing stances and a judge selects rational outcomes (Zhang et al., 2024b), and reflection-based methods such as Reflexion (Shinn et al., 2023) and COPPER (Bo et al., 2024), which refine decisions through self-critique or counterfactual rewards. These approaches highlight the value of reasoning over alternatives but remain limited to single-agent self-reflection or debate.

**Multi-LLM Agent Collaborative Intelligence.** Recent work argues that progress toward AGI hinges less on scaling monolithic models and more on enabling coordinated intelligence among specialized LLM agents. MACI (Chang, 2025a) formalizes this view, showing how persistent memory, role-specific reasoning, and deliberative dialogue unlock system-level capabilities. A series of systems instantiate this paradigm: CRIT (Chang, 2023) probes argument robustness through counterfactual reasoing, SocraSynth (Chang, 2024b) extends this paradigm to multi-agent settings via rubric-guided dialogue; EVINCE (Chang, 2024a) grounds multi-agent dialogue in retrieved evidence, optimizing inter-agent communication; ALAS (Geng & Chang, 2025) introduces disruption-aware planning via structured compensation policies to repair failed multi-agent plans, SagaLLM (Chang & Geng, 2025) extends this into a full transactional framework for multi-agent LLM planning; and Checks-and-Balances (Chang, 2025b) decomposes ethical deliberation into specialized roles. These works together illustrate the emerging principles behind reasoning, coordination, planning, and alignment in multi-LLM systems.

EvoCF operates alongside existing multi-LLM collaborative frameworks while focusing on the execution layer of embodied multi-agent tasks. In these settings, plan reliability depends on capturing task preconditions, feasibility, and coordination constraints under partial observability, factors that cannot be resolved through dialogue-based refinement alone. To address this gap, EvoCF applies a memory-driven counterfactual planning process that proposes and evaluates alternative joint actions grounded in prior execution traces. Its symbolic constraint library self-evolves by continually extracting and updating task, spatial, and coordination rules from new failures. This evolving rule base enables EvoCF to iteratively refine and improve joint plans, resulting in strategies that are both feasible and robust in complex embodied environments. Taken together, EvoCF enriches the MACI family with a path toward agents that continually refine their joint plans through counterfactual exploration and memory-guided evaluation grounded in accumulated experience.

## 3 METHODOLOGY

### 3.1 MEMORY-GUIDED MULTI-AGENT COLLABORATION PROBLEM

**Problem Setting.** We study embodied collaborative planning in partially observable environments, and formulate the problem as a multi-agent partially observable Markov decision processes (POMDPs) $\mathcal{E} = \langle \mathcal{S}, \{\mathcal{O}_i\}_{i=1}^N, \{\mathcal{A}_i\}_{i=1}^N, \mathcal{G} \rangle$. $\mathcal{S}$ is the state space, $\mathcal{O}_i$ is the observational space of agent $i$, $\mathcal{A}_i$ is the action space of agent $i$, and $N$ is the number of agents. At timestep $t$, each agent receives a partial observation $o_t^i$ and they must coordinate with each other to achieve the shared goal $\mathcal{G}$.

Given a high-level instruction $\tau$ for task $\mathcal{G}$, the planner must synthesize suitable actions for each agent. This process unfolds as: (i) decomposing $\mathcal{G}$ into a sequence of subgoals $\{g_1, g_2, \ldots, g_K\}$, (ii) grounding each subgoal into joint actions $(a_t^1, \ldots, a_t^N)$, and (iii) coordinating execution under constraints such as temporal ordering, role specialization, and resource sharing. The core challenge is to ensure that the induced joint policy $\pi = (\pi^1, \ldots, \pi^N)$ produces strategies that are not only executable but also robust to coordination failures and environmental uncertainty.

**Modular Planning Roles.** As shown in Figure 1, our framework builds upon the modular multi-agent planning pipeline from LLaMAR (Nayak et al., 2024), which organizes multi-agent collaboration as a sequential process of task decomposition and action allocation among agents. **EvoCF** extends the pipeline with three instruction-guided modules that automate counterfactual planning and strengthen multi-agent collaboration: (i) the **Counterfactual Plan Generator**, which introduces evolutionary operators to explore diverse constraint-guided alternatives; (ii) the **Retrieval-Augmented Counterfactual Evaluator**, which grounds these candidates in past outcomes and symbolic constraints to assess their viability; and (iii) the **Symbolic Constraint Inductor**, which distills failure patterns into reusable rules that accumulate in memory. Appendix A provides the prompt details of EvoCF.

## 3.2 Experience-Grounded Discovery of Symbolic Constraints

In this section, we introduce our rule generation method, which forms the core mechanism enabling effective counterfactual action generation and evaluation. The central challenge lies in systematically transforming knowledge derived from failure signals and task descriptions into a form that can guide reasoning. To address this, we propose a structured memory representation coupled with symbolic constraint construction, providing a principled way to capture, organize, and operationalize such knowledge for adaptive and self-evolving planning.

**Structured Memory.** We maintain a structured, cross-trial memory $\mathcal{M}$ that records outcome-grounded experience at the transition level. $\mathcal{M}$ serves as the central knowledge pool for efficient downstream retrieval, induction and reasoning. Formally, each record is $m = \left( \mathbf{o}_t, \mathbf{a}_t, \{\psi_t^k\}_{k=1}^M \right)$, where $\mathbf{o}_t = (o_t^1, \ldots, o_t^N)$ are agents' consecutive partially-observed states, $\mathbf{a}_t = (a_t^1, \ldots, a_t^N)$ are the joint actions, and $\{\psi_t^k\}_{k=1}^{K_t}$ is a set of *typed structural annotations* that provide multi-perspective interpretations of the experience. We use lowercase symbols for realized quantities ($\mathbf{o}_t, \mathbf{a}_t$) and uppercase for sets/types ($\mathcal{M}, \Psi$). Concretely, each $\psi^k \in \Psi$ may encode one of the following facets:

$$\psi^k \in \left\{ \begin{array}{ll} \psi^{\mathrm{plan}} \text{ (subgoal id, role assignment);} & \psi^{\mathrm{eff}} \text{ (observed effects);} \\ \psi^{\mathrm{pre}} \text{ (preconditions);} & \psi^{\mathrm{fail}} \text{ (failure codes, induced rule fragments)} \end{array} \right\}.$$

Each structural element $m \in \mathcal{M}$ is annotated with calibrated metadata (e.g., confidence scores, support counts), enabling principled aggregation across experiences. This typed design transforms $\mathcal{M}$ into an *experience-grounded structured library* rather than a flat trajectory buffer, and aligns its reasoning surface with the multi-agent POMDP: preconditions and effects over $\{\mathcal{O}_i\}_{i=1}^N$, joint actions over $\{\mathcal{A}_i\}_{i=1}^N$, and role/temporal dependencies across agents.

We formalize retrieval through a compositional query operator $f_{\mathrm{query}} : q \times \mathcal{M} \mapsto \Psi^*$, that maps a query and the memory into an aggregated set of structural facets $\Psi^* \subset \Psi$. Crucially, this operator provides a principled mechanism to retrieve, compose, and reason over counterfactual outcomes, induce symbolic constraints, and generalize coordination patterns, thereby enabling robust multi-agent planning beyond single-pass LLM outputs.

**Symbolic Constraint Induction.** We introduce a symbolic constraint induction mechanism that discovers and continually refines reusable rules from experiences and failure signals in multi-agent collaboration. The induced rules expand dynamically with new interactions, forming an evolving set that grounds counterfactual planning in structured knowledge and enables the discovery of robust collaborative strategies.

Formally, we define the set of symbolic constraints as $\mathcal{R}$, partitioned into two complementary categories that correspond to the dimensions of plan evaluation:

- **Task-Dependency Constraints** ($\mathcal{R}_{\mathrm{dep}}$): capture structural preconditions and causal/temporal effects that govern task feasibility independent of collaboration. These constraints align with *single-agent execution metrics* $M_{\mathrm{SA}}$, such as `ActionValidity`, `ObjectReachability`, and `SpatialFeasibility`, ensuring each agent's actions are executable and consistent with physical and semantic conditions.
- **Multi-Agent Coordination Constraints** ($\mathcal{R}_{\mathrm{coord}}$): capture interaction-specific requirements that emerge only in collaborative settings. These constraints correspond to *multi-agent coordination metrics* $M_{\mathrm{MA}}$, such as `TemporalConsistency`, `LoadBalance`, and

Figure 1: Overview of the EvoCF framework for **memory-guided evolutionary counterfactual planning**. Given an initial multi-agent action plan, EvoCF explores counterfactual alternatives through **evolutionary mutation operators** guided by symbolic rules distilled from past failures. Each candidate plan is then evaluated by a **experience-guided evaluator** module that queries structured memory for relevant outcomes and constraints, to select the best one under multi-objective criteria. By integrating symbolic rule induction, evolutionary plan search, and experience-driven evaluation, EvoCF yields more robust and adaptive multi-agent collaboration strategies.

    `GoalAlignment`, ensuring effective synchronization, conflict avoidance, and cooperative goal satisfaction.

These two categories together define the symbolic rule space: $\mathcal{R} = \mathcal{R}_{\text{dep}} \cup \mathcal{R}_{\text{coord}}$, providing a principled decomposition of constraints that bridges individual feasibility with collective coordination in multi-agent planning. We summarize induced constraints across six metrics in Appendix B.3.

Constraints are induced directly from memory records $m \in \mathcal{M}$ via a generator: $\mathcal{C}_{\text{gen}} : m \mapsto \{r_1, \dots, r_k\}$ that maps each failure-annotated transition to a set of candidate rules. Each rule is annotated as $r = (\phi_r, \tau_r, \mu_r)$, where $\phi_r$ is a symbolic formula (preconditions, effects, or coordination logic), $\tau_r \in \{\text{dep}, \text{coord}\}$ is its category, and $\mu_r$ stores calibrated metadata (e.g., confidence, support). The rule set evolves online via a simple update $\mathcal{R} \leftarrow \text{Dedup}\left(\mathcal{R} \cup \mathcal{C}_{\text{gen}}(m)\right)$ ensuring compactness and continual refinement as new experiences arrive.

These induced rules offer a human-interpretable substrate that grounds counterfactual reasoning, significantly enhancing agents' ability to anticipate outcomes and coordinate effectively in complex multi-agent settings.

### 3.3 EVOLUTIONARY COUNTERFACTUAL PLANNING

**Compositional Experience Retrieval.** We introduce a compositional experience retrieval framework over the structured memory described in Sec. 3.2. A *compositional query* $q$ flexibly encodes different combinations of contextual signals, such as the current observation, a candidate action, or a subgoal tuple, optionally fused with structural identifiers (e.g., memory index or semantic tag). Concretely, we denote $q = \text{AGG}\left((x_1, x_2, \dots, x_k), \xi\right)$, where $\text{AGG}(\cdot)$ aggregates contextual tuples and $\xi$ encodes identity information. This enables retrieval at multiple levels of granularity, ensuring that both local context and higher-order dependencies are grounded in past experience. Formally, the universal retrieval function over the structured memory is defined as:

$$R(q) = \bigcup_{m \in \mathcal{N}(q)} \Psi^{\text{rule}}(m), \qquad \mathcal{N}(q) = \text{Top-k}\left\{m \in \mathcal{M} \,\Big|\, \frac{\mathbf{e}(q) \cdot \mathbf{e}(m)}{\|\mathbf{e}(q)\| \, \|\mathbf{e}(m)\|}\right\}, \qquad (1)$$

where $\mathbf{e}(\cdot)$ is the embedding function, $\mathcal{M}$ is episodic memory, and $\Psi^{\text{rule}}(m)$ extracts symbolic rules associated with memory $m$. This compositional formulation provides a principled mechanism to *learn from prior outcomes and induced constraints*, while offering a unified interface for both counterfactual plan generation and evaluation.

**Counterfactual Plan Generation.** EvoCF drives multi-agent collaboration beyond fixed plans by *evolving* them under symbolic and experience-driven constraints. At its core, counterfactual planning is cast as a *guided evolutionary process* that balances diversity with feasibility, ensuring candidate plans remain both task-relevant and plausibly correct. The procedure unfolds in three key steps:

(i) **Gene Representation:** A joint plan for $N$ agents at timestep $t$ is treated as an *individual*, where each agent's assigned action is modeled as a *gene*, i.e., a *receptacle* for symbolic content that can be perturbed by evolutionary operators:

$$P_t = \langle a_t^1, a_t^2, \ldots, a_t^N \rangle, \qquad \texttt{gene}: a_t^i \mapsto \texttt{Act}(\texttt{agent}^i, \texttt{object}). \qquad (2)$$

Each gene $a_t^i$ encodes an agent–action binding (e.g., $\texttt{NavigateTo(Alice, Fridge)}$), while the sequence $P_t$ captures the full multi-agent plan. This representation serves as a substrate for mutation operators, enabling instruction-following exploration of counterfactual plans while respecting symbolic and experience-driven constraints.

(ii) **Mutation Operators:** EvoCF explores counterfactual diversity through a discrete set of symbolic mutation operators,

$$\Omega = \{\texttt{SwapAgent, InsertAction, DeleteAction, ReplaceObject}\},$$

where each $\omega \in \Omega$ perturbs the gene sequence $P_t$ by altering agent–action bindings in a structured manner. Unlike blind perturbations, these operators are *guided* by retrieved experience: given the current observation $\mathbf{o}_t$, the compositional retrieval in Eq. 1 returns a neighborhood of relevant memory items $\{m\}$, together with their symbolic rules $\mathcal{R}(\mathbf{o}_t)$ and outcome traces $\{\psi^{\text{out}}(m)\}$. This context steers mutations toward plausible alternatives, e.g., inserting an action known to resolve past failures, or replacing an object in line with affordances observed in similar scenes.

Formally, the counterfactual plan space is defined as:

$$\mathcal{P}_{\text{cf}}(P_t, \mathbf{o}_t) = \left\{ f_{\text{plan}}\big(P_t, \omega, \mathcal{R}(\mathbf{o}_t), \{\psi^{\text{out}}(m)\}\big) \,\Big|\, \omega \in \Omega \right\}, \qquad (3)$$

where $f_{\text{plan}}$ denotes a reasoning function that conditions on retrieved constraints and outcomes to guide operator application. This design ensures that mutations remain both *diverse* and *task-relevant*, rather than arbitrary, allowing EvoCF to explore counterfactuals that better reflect feasible multi-agent collaboration strategies.

**Counterfactual Plan Evaluation.** EvoCF intuitively accesses counterfactual outcomes through the compositional retrieval mechanism (Eq. 1), constructing a *transition-level evidence base* for evaluation. Specifically, it forms compositional queries over local state–action tuples $(\mathbf{o}_t, \mathbf{a}_t)$ and retrieves past outcomes $(\mathbf{o}_{t+1})$ together with symbolic constraints $\{\psi^{\text{rule}}(m)\}$. This joint evidence anchors the evaluation of each counterfactual plan in concrete experiences while ensuring consistency with induced rules.

EvoCF employs an experience-driven, constraint-guided reasoning function that conditions the fitness of each candidate on the current context, retrieved outcomes, and symbolic rules. Through this reasoning process, candidates $P'$ are ultimately ranked:

$$\text{Rank}(P') \propto f_{\text{eval}}\big(P', \mathbf{o}_t, \mathcal{R}(\mathbf{o}_t, \mathbf{a}_t), \{\psi^{\text{out}}(m)\}\big), \qquad P' \in \mathcal{P}_{\text{cf}}. \qquad (4)$$

Internally, $f_{\text{eval}}$ functions as a reasoning layer that integrates retrieved evidence and induced constraints to mimic multi-objective aggregation, with rule confidences modulating the relative emphasis on feasibility, progress, and coordination. The formulation is invariant under any strictly monotone transformation of $f_{\text{eval}}$, so only the induced ordering of candidate plans is relevant. Intuitively, this yields a *world-model-free reasoning layer*: rather than forward-simulating dynamics, EvoCF leverages structured memory to anticipate consequences and eliminate implausible strategies. To the best of our knowledge, EvoCF is the first framework that unifies evolutionary counterfactual planning with retrieval-augmented reasoning, offering evaluations that are simultaneously memory-grounded, symbolically constrained, and interpretable.

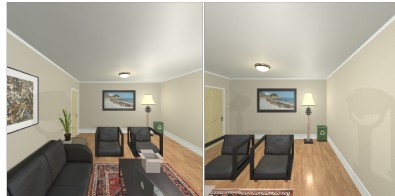

(a) Kitchen           (b) LivingRoom

Figure 2: Photorealistic rendering of household scenarios in the MAP-THOR simulator enables the usage of multiple autonomous robots to carry out daily tasks. (a) and (b) show the egocentric views of two agents operating in the kitchen and living room.

## 4 EXPERIMENTS

To evaluate the effectiveness of our method in improving multi-agent collaboration under long-horizon and partially observable settings, we build upon the experimental framework introduced in Nayak et al. (2024). Specifically, we adopt the MAP-THOR (Nayak et al.) benchmark suite, a diverse and challenging collection of embodied household tasks based on the AI2-THOR (Kolve et al., 2017) simulator with a multi-agent setup. As shown in Figure 2, all the experiments were performed in the single-room floor plans. For consistent and fair comparison, we adopt LLaMAR's experimental setup and evaluation protocol. Concretely, we evaluate on 45 tasks in MAP-THOR, each instantiated with five room layouts spanning explicit to underspecified goals. We retain the original baselines, including Act, Chain-of-Thought (Wei et al., 2022), ReAct (Yao et al., 2022), SmartLLM (Kannan et al., 2024), and CoELA (Zhang et al., 2024a), and report results using the same metrics defined in Nayak et al. (2024):

- **Success Rate (SR):** Fraction of episodes where all subtasks are successfully completed.
- **Transport Rate (TR):** Fraction of subtasks completed within an episode.
- **Coverage (C):** Fraction of successful interactions with target objects.
- **Balance (B):** Ratio of min/max successful high-level actions across agents, reflecting collaboration equity.
- **Average Steps (L):** Number of high-level steps taken before completion or timeout.

While the values in baseline results are obtained with GPT-4V (now deprecated), our experiments use GPT-4o, the latest vision-language model from OpenAI, under identical simulation conditions to ensure fair and reproducible evaluation.

| Methods | SR↑ | TR↑ | C↑ | B↑ | L↓ |
|---|---|---|---|---|---|
| Act | 0.33 | 0.67 | 0.91 | 0.59 | 24.92 |
| ReAct | 0.34 | 0.72 | 0.92 | 0.67 | 24.08 |
| CoT | 0.14 | 0.59 | 0.87 | 0.62 | 28.40 |
| SmartLLM | 0.11 | 0.23 | 0.91 | 0.45 | 29.87 |
| CoELA | 0.25 | 0.46 | 0.76 | 0.73 | 28.93 |
| LLaMAR | 0.66 | 0.91 | 0.97 | 0.82 | 21.87 |
| EvoCF (ours) | **0.84** | **0.95** | **0.99** | **0.89** | **18.69** |

Table 1: Comparison of evaluation metrics against baselines averaged across all tasks for the 2-agent MAP-THOR scenarios.

### 4.1 RESULTS AND DISCUSSION

Table 1 compares our method, LLaMAR, with other baselines in a 2-agent scenario. EvoCF achieves consistent and substantial gains across all evaluation metrics, outperforming strong baselines such as LLaMAR, ReAct, and CoELA. In particular, EvoCF attains a success rate of 0.84, surpassing LLaMAR by 18%, while also improving transport rate (0.95) and coverage (0.99), indicating that agents not only complete more subtasks but also interact more reliably with task-relevant objects. These improvements are driven by EvoCF's symbolic rule induction mechanism, which extracts reusable constraints from past failures, covering both single-agent feasibility (e.g., object accessibility, action

|  | SwapAgent | InsertAction | DeleteAction | ReplaceObject |
|---|---|---|---|---|
| Occurrence Frequency | 12% | 39% | 22% | 27% |
| Adoption Frequency | 8% | 17% | 8% | 10% |

Table 2: **Occurrence Frequency** refers to the percentage of generated counterfactual plans that include each mutation operator; **Adoption Frequency** refers to the proportion of counterfactual plans counterfactual plans produced by each operator are selected for execution, where 57% triggers original plan. The results show that the counterfactual planning replaces original plan with a noticeable high chance of 43%, highlighting the crucialness of counterfactual mutation in improving multi-agent planning robustness and flexibility.

preconditions) and multi-agent coordination dependencies (e.g., spatial conflicts, role interference). These rules serve as essential structural priors that guide the generation of counterfactual plans under physical and social constraints.

The evolutionary counterfactual planner leverages these symbolic constraints to explore alternative action assignments and subgoal orderings under the current observation, enabling the discovery of robust execution paths that preemptively avoid likely failure points. Meanwhile, the experience-driven evaluator further filters these candi-

| # Agents | SR↑ | TR↑ | C↑ | B↑ | L↓ |
|---|---|---|---|---|---|
| 2 | 0.84 | 0.95 | 0.99 | 0.89 | 18.69 |
| 3 | 0.87 | 0.96 | 0.99 | 0.81 | 17.36 |
| 4 | 0.82 | 0.93 | 0.99 | 0.74 | 19.51 |

Table 3: Benchmarking EvoCF with different numbers of agents. EvoCF exhibits stable success and coordination despite scaling complexity.

dates by retrieving relevant prior experiences and evaluating their long-term viability, not just based on current feasibility, but also on whether the plan aligns with patterns that have historically led to successful completions. This combination prevents overfitting to short-horizon fixes and promotes globally coherent strategies. These capabilities contribute to the observed improvements in balance (0.89), as agents are more equitably involved in plan execution, and the reduction in average steps (18.69), as failure-prone branches are pruned early in planning. These results highlight EvoCF's capacity to integrate structured symbolic reasoning with experiential generalization for more reliable and efficient multi-agent collaboration.

To assess the relative impact of each mutation operator on plan quality and selection, we analyzed their usage frequency during counterfactual generation and their contribution to top-ranked plans. As shown in Table 2, these results show that InsertAction is the most impactful operator: it appears in 39% of generated plans and contributes to 17% of selected top-ranked plans. This aligns with the nature of MAP-THOR tasks, which often require fine-grained spatial adjustments (e.g., moving slightly, rotating to align) that are not captured in the initial plan. InsertAction enables such refinements by adding subtle but necessary corrections. ReplaceObject and DeleteAction provide moderate contributions, useful for recovering from object choice errors or resolving collisions. SwapAgent is less frequently used, as task roles are typically well-decomposed in the initial plan.

To assess EvoCF's scalability, we evaluate its performance under increasing numbers of agents in the same environment. As shown in Table 3, EvoCF demonstrates improved success rate and reduced planning steps when scaling from two to three

|  | K=1 | K=3 | K=5 | K=10 |
|---|---|---|---|---|
| SR↑ | 0.77 | 0.81 | 0.84 | 0.82 |

Table 4: Sensitivity analysis on the depth of retrieved trajectories $K$ used by the evaluator.

agents, indicating effective utilization of additional agent capacity through symbolic coordination. However, when scaling to four agents, performance slightly declines: the success rate drops modestly and load balance deteriorates, reflecting increased coordination complexity. This is largely due to agent congestion and interaction overhead in shared environments, which lead to more retries and blocked plans. Additionally, the MAP-THOR benchmark enforces a fixed high-level planning step cap (L=30), which constrains the per-agent planning horizon as the team size grows. This results in more truncated or incomplete subtask plans, rather than failures from constraint retrieval or computational limits.

|  | SR↑ | TR↑ | C↑ | B↑ | L↓ |
|---|---|---|---|---|---|
| EvoCF (w/ Random CF Generator) | 0.68 | 0.91 | 0.97 | 0.83 | 21.37 |
| EvoCF (w/ Heuristic Evaluator) | 0.72 | 0.92 | 0.98 | 0.85 | 19.83 |
| EvoCF (w/o Rule Induction) | 0.75 | 0.93 | 0.98 | 0.87 | 19.62 |
| **EvoCF** | **0.84** | **0.95** | **0.99** | **0.89** | **18.69** |

Table 5: Ablation study on the impact of counterfactual plan generation, experience-driven evaluation, and Rule Induction. All metrics are evaluated on the final selected plan for each episode.

To assess the impact of memory retrieval depth on evaluator performance, we conducted a sensitivity analysis by varying the number of retrieved trajectories $K \in \{1, 3, 5, 10\}$, while keeping all other components fixed. As shown in Table 4, increasing $K$ from 1 to 5 leads to steady improvements in success rate, as the evaluator benefits from more diverse and informative past cases. However, performance slightly drops at $K = 10$, suggesting that additional trajectories may introduce redundancy or noise. These results indicate that $K = 5$ achieves a good balance between relevance and diversity in retrieved memory, and is used as the default setting in all experiments.

We conduct an ablation study to assess the impact of three key modules in EvoCF. As shown in Table 5, using random mutations results in marginal improvement over LLaMAR, as the evaluator often falls back to the original plan when most counterfactuals are low-quality. In contrast, removing the experience-driven evaluator and relying on heuristics yields better performance by selecting from structurally valid candidates, but still underperforms the full system due to lack of experience-grounded judgment. Further, ablating rule induction removes structural guidance for mutation and deprives the evaluator of reliable constraints, resulting in less targeted counterfactual plan generation and weaker plan selection. These results confirm that all three components are critical: mutation operators enable the generation of diverse and feasible counterfactuals; experience-driven evaluation ensures robust and context-aware plan selection; and rule induction extracts generalizable structural priors from past failures to guide future reasoning. Additionally, we present a reusable rule constraint case across tasks in Appendix B.1.

## 4.2 CASE STUDY

We present a case study on the **Turn on all stove knobs** task in MAP-THOR, where agents Alice and Bob collaboratively explore and operate stove knobs throughout the kitchen scenario. As shown in Figure 3, we illustrate how EvoCF improves multi-agent collaboration via symbolic rule induction, counterfactual plan search, and memory-guided evaluation.

**Symbolic Rules Induction.** We demonstrate how EvoCF improves multi-agent planning by analyzing Step 10 with subtask *Open All Drawers*. The plans at Step 9 are: `[NavigateTo(Stove_2), OpenObject(Stove_1)]`. Execution feedback shows that Alice failed to reach `Stove_2` due to obstruction, while Bob's open attempt failed because he was not properly aligned with `Stove_2`. EvoCF captures this transition in memory and annotates it with structured outcomes. From the failure signals, it induces the following reusable symbolic constraints:

- `NavigateTo(?Object) → Avoid(CollisionWith(?Agent))`
- `OpenObject(?Object) → Facing(?Object)`

These constraints reflect spatial feasibility and interaction prerequisites that were violated in this step. Once stored in the evolving rule library, they serve as useful prior to guide future plans.

**Counterfactual Plan Generation.** The planner generates an **Original Plan** (`[Move(ahead), OpenObject(Stove_1)]`) in Step 6 that avoid the prior errors. The counterfactual plan generator mutates the original plan under the above constraints to generate alternative joint plans. For Step 6, it proposes the following counterfactual variants:

- **CF Plan A:** `[Rotate(30), Move(left)]` — Alice rotates to another angle for reduce collision risk, Bob moves left to get closer to and face the target.
- **CF Plan B:** `[Idle, Move(left)]` — Alice pauses, Bob retries the open action with another object.

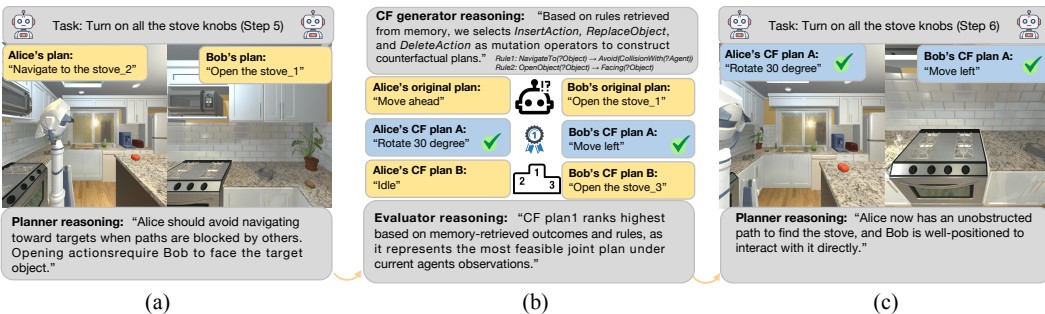

(a)  (b)  (c)

Figure 3: Illustration of EvoCF's reasoning process on the *Turn on all stove knobs* task. (a) The original plan at Step 5 fails: Alice attempts to navigate toward *stove_2* but is blocked, and Bob fails to open *stove_1* due to misalignment. (b) Guided by the memory-retrieved rules, EvoCF generates multiple counterfactual plans and evaluates them by an evaluator. *CF Plan A* is selected as the top-ranked plan to interact with the environment.(c) After executing *CF Plan A*, the agents are in favorable states: Alice has an unobstructed path for further exploration, and Bob is well-positioned to interact with the stove knob, enabling successful task completion.

Each counterfactual plan applies specific mutation operators based on the retried symbolic constraints: CF Plan A applies `InsertAction` to both Alice's and Bob's actions. CF Plan B applies `DeleteAction` to Alice's action and `InsertAction` to Bob's action.

**Memory-guided Evaluation.** Rather than scoring plans heuristically, EvoCF retrieves past transitions with similar object locations and interaction failures. For example, prior failures of `OpenObject` due to misalignment support the constraint `Facing(Object)`, while collision cases inform the need for repositioning before navigation. EvoCF then ranks candidate plans by combining symbolic constraints with experience-grounded outcomes. At Step 6, **CF Plan A**, `[Rotate(30), Move(left)]`,is selected as most promising. This plan outperforms the original and other variants by jointly resolving spatial interference and misalignment: Alice takes a clearer path more aligned with the task goal, while Bob adjusts orientation, improving execution success and subgoal coordination. This case illustrates how EvoCF derives effective, failure-aware collaboration strategies through its integrated planning pipeline.

# 5 LIMITATIONS

While EvoCF demonstrates strong promising performance, several components admit further generalization. (1) *Mutation Operators:* Our symbolic mutation operators are manually designed for common plan-editing patterns. As shown in ablation, their effectiveness degrades under random selection, suggesting limited generalization and motivating learning-based strategies for improved adaptability. (2) *Symbolic Constraints:* Integrating more open-ended knowledge or learning from broader experience could enhance generality. (3) *Evaluator Design:* Beyond memory retrieval, a learned world model could enable rollout-based verification of counterfactual plans.

# 6 CONCLUSION

In this paper, we introduce a framework for multi-agent collaboration via memory-guided evolutionary counterfactual planning. Built on structured experience memory, our system synthesizes symbolic constraints from past interactions and retrieves them compositionally to guide both the generation and evaluation of counterfactual plans. This process enriches the diversity of meaningful plan candidates and enables agents to proactively revise failures into more coordinated strategies. Empirical results across multi-agents collaboration benchmarks show significant gains in success rate, coordination, and efficiency. These findings demonstrate that counterfactual plan search offers a principled path toward more robust, adaptive, and collaborative multi-agent behaviors.

Our future work will extend counterfactual reasoning from action assignments to subtask-level planning, enabling finer-grained edits such as goal reordering, subgoal decomposition, and partial plan substitution. We also aim to develop more expressive evaluation mechanisms, including outcome verification and deliberative agents capable of self-reflection and multi-step plan refinement, to further improve the reliability and adaptability of collaborative planning.

## ETHICS STATEMENT

This work adheres to the ICLR Code of Ethics. Our study focuses on embodied multi-agent collaboration in simulated environments (MAP-THOR) and does not involve human subjects or sensitive personal data. The proposed methods are evaluated entirely within synthetic scenarios, thereby avoiding risks related to privacy or real-world safety. We acknowledge that embodied AI techniques may have downstream societal impacts if deployed in real-world contexts, such as automation in domestic or industrial settings. We emphasize that our contributions are intended for research purposes and should be carefully assessed before any real-world application. No conflicts of interest or external sponsorship influenced the reported results.

## REPRODUCIBILITY STATEMENT

We have taken deliberate steps to ensure the reproducibility of our results. All experiments are based on the MAP-THOR benchmark, which is publicly available. Detailed descriptions of experimental setups, evaluation metrics, and baselines are provided in Section 4, with implementation details and prompts documented in Appendix A. Anonymized source code and configuration scripts are provided as part of the submission to allow independent verification of results.

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

APPENDIX

# A  PROMPTING DETAILS

We describe the prompts used for each of the modules used in EvoCF:

---

**Symbolic Constraint Inductor**

```
<inputs>
    {subtask}
    {observation}
    {action}
    {failure_reason}
</inputs>
```

You are a symbolic reasoning agent helping a team of {num_agents} embodied agents improve their multi-agent collaboration and planning by learning from individual failures.

Your task is to analyze one failure experience and induce symbolic constraints that, if satisfied, would have prevented the failure. These symbolic constraints help guide future planning and execution, and should be categorized into one of the following six types:

# Constraint Type Options:
1. Multi-Agent Coordination Metrics:
**Temporal Consistency**: Ensure agents follow proper semantic action order.
**Load Balance**: Avoid redundancy or idleness across agents.
**Goal Alignment**: Ensure agents' goals are mutually compatible.

2. Single-Agent Execution Metrics:
**Action Validity**: Action must align with the agent's current state.
**Object Reachability**: The object/location must be spatially reachable.
**Spatial Feasibility**: The action must be physically possible, not blocked.

## Output must follow this format:
```
{
symbolic_rule: <a logic-style symbolic expression capturing the structural requirement.>,
description: < a plain natural language explanation of the rule.>,
type: <the option of constraint>
}
```

---

**Counterfactual Plan Generator**

```
<inputs>
    {subtask}
    {observation}
    {original_plan}
    {constraints}
</inputs>
```

You are a counterfactual plan generator for a team of {num_agents} embodied multi-agent robots collaboration.

Your goal is to propose alternative multi-agent joint action plans based on the current timestep's context. These plans are meant to explore meaningful variations over the original plan, guided by symbolic constraints and past failure experiences.

You will be given:
**Subtasks**: The current goal or sub-goals assigned to each agent.
**Original plan**: The current joint plan consisting of one action per agent.
**Observations**: The latest environmental observations available to each agent.
**Constraints**: A set of symbolic constraint rules retrieved from past failure cases.

You are only allowed to select action from the following predefined robot action space:

["navigate to object $< object\_id >$", "rotate in $< rotation >$ direction", "pick up object $< object\_id >$", "put object on $< receptacle_id >$", "open object $< object\_id >$", "close object $< object\_id >$", "slice object $< object\_id >$", "toggle object $< object\_id >$ on", "toggle object $< object\_id >$ off", "clean object $< object\_id >$", "look up by angle $< angle >$", "look down by angle $< angle >$", "move in $< translation >$ direction", "stay idle"]

Here, "stay idle" is used when you want the robot to stay idle for one time step. This could be used to wait for the other robot to complete its subtask. Use it only when you think it is necessary. Here $< rotation >$ can be one of ["Right", "Left"]. Here $< angle >$ is the angle in degrees and can only be one of [30, 60, 90, 120, 150, 180]. Here $< translation >$ can be one of ["Ahead", "Back", "Left", "Right"].

You must generate **N counterfactual joint action plans**, where each plan is a variation of the current original plan 'A' created by applying **one or more mutation operators** to one or more agents. The mutation operators available to you include:

---

- **SwapAgent**: Swap the actions assigned to agents.
- **InsertAction**: Replace the action verb, keeping the object the same.
- **DeleteAction**: Set an agent to do nothing at this timestep (stay idle).
- **ReplaceObjective**: Replace the object argument of the action, keeping the verb the same.

You must:
- Use retrieved constraint rules and agent current observations to infer which agent actions are likely to lead to constraint violations or failure, based on similar past failure experiences.
- For the action(s) identified above, select appropriate mutation operators to explore counterfactual alternatives that could avoid these failures or improve success likelihood.  - Ensure that each generated plan remains **semantically consistent** with the subtask.
- Ensure that each action is from the available action space listed above.
- Avoid generating plans that violate known constraints unless explicitly exploring failure.

## Format your output as a list of candidate plans:
{
1. $[< Action >, < Action >, ..., < Action >]$,
...
N. $[< Action >, < Action >, ..., < Action >]$
}

---

**Experience-Driven Evaluator**

```
<inputs>
   {subtask}
   {observation}
   {candidate_plans}
   {retrieved_constraints}
   {retrieved_outcomes}
</inputs>
```

You are a collaborative reasoning assistant helping a team of {num agents} embodied robots evaluate possible collaboration strategies at the current time step.

You are given:
**Subtasks**: The current high-level subtask.
**Observations**: Each agent's local observation.
**Candidate plans**: A list of candidate joint plans, each consisting of one action per agent.
**Constraints**: A set of symbolic rules retrieved from memory.
**Outcomes**: A set of outcomes that provide multi-perspective interpretations of the experiences from similar situations.

Your goal is to **reason over the retrieved symbolic rules and outcomes**, and **select the most promising plan** for the current context. **You must not assign numerical scores or simulate the environment**. Instead, base your decision on:
- Which plan **best satisfies the retrieved symbolic constraints**?
- Which plan **avoids failure patterns seen in retrieved outcomes**?
- Which plan is **most likely to lead to progress** based on past experiences?

You should:
- Use multi-step reasoning to analyze how each plan aligns (or conflicts) with the constraints and past outcomes.
- Consider both **single-agent feasibility** and **multi-agent coordination**.
- Highlight which constraints each plan violates or satisfies.
- Rank the plans purely by reasoning and justify your choice.

## Output must follow this format:
{
"plan_ranking": [
{
"plan": $[< Action >, ..., < Action >]$,
"reasoning": " $< explanation >$ "
},
...]
}

---

# B  ADDITIONAL EXPERIMENT RESULTS

## B.1  SYMBOLIC CONSTRAINT TRANSFER CASE STUDY

To isolate the reusability of symbolic constraints across tasks, we design a minimal intervention experiment: we induce constraints from a single source task (`put bread, lettuce, and a tomato in the fridge`), and inject them as structured planning priors when solving unrelated tasks (`put the pots and pans on the stove burners, open all drawers`).

These constraints are provided to the planner without invoking the counterfactual generator or evaluator, and no new rules are induced for the test tasks. Thus, any improvement in task completion indicates that the original constraints encode transferable physical and coordination knowledge applicable across tasks.

As shown in Table 6, injecting transferred constraints consistently improves task performance across all key metrics. For instance, the success rate on put the pots and pans on the stove increases from 0.60 to 0.80, and on open all drawers from 0.40 to 0.60. Transport rate, collaboration balance, and average steps also improve, with minimal or no degradation in coverage. These improvements arise despite the constraints being induced from a structurally different task, supporting the hypothesis that EvoCF induces structurally generalizable rules.

This case study confirms that EvoCF's symbolic constraints encode transferable knowledge about action feasibility and multi-agent coordination. Even when applied to unseen tasks with different object semantics and room layouts, the rules provide effective priors for generating more executable and balanced joint plans, highlighting their potential for reuse in continual and multi-task embodied planning.

| Target Task | Setting | SR ↑ | TR ↑ | C ↑ | B ↑ | L ↓ |
|---|---|---|---|---|---|---|
| Put the pots and pans on the stove | Planner Only | 0.60 | 0.88 | 0.95 | 0.80 | 20.13 |
| | w/ Transferred Constraints | 0.80 | 0.93 | 0.97 | 0.82 | 18.55 |
| Open all drawers | Planner Only | 0.40 | 0.75 | 0.94 | 0.75 | 22.41 |
| | w/ Transferred Constraints | 0.60 | 0.88 | 0.94 | 0.79 | 20.95 |

Table 6: Evaluating the transferability of symbolic constraints induced from the *put bread, lettuce, and tomato in the fridge* task. Constraints are injected into the planner for new tasks without using counterfactual plan generation or evaluation. Improved planning performance would indicate that symbolic constraints are reusable across tasks.

## B.2 COMPUTATION OVERHEAD ANALYSIS

As shown in table7 and table8, we report the per-module latency and token cost of EvoCF compared to the LLaMAR baseline.

| # Latency | LLaMAR-Moudles | Inductor | CF Generator | Evaluator | Toal |
|---|---|---|---|---|---|
| LLaMAR | 7.61 | - | - | - | 166.43 |
| EvoCF (ours) | 7.61 | 1.46 | 2.17 | 2.31 | 269.76 |

Table 7: Average per-step latency (seconds) by module and total runtime per episode.

| # Token | LLaMAR-Moudles | Inductor | CF Generator | Evaluator | Toal |
|---|---|---|---|---|---|
| LLaMAR | 1.58K | - | - | - | 35.65K |
| EvoCF (ours) | 1.58K | 2.45K | 4.27K | 5.12K | 47.82K |

Table 8: Average per-step token usage by module and cumulative total per episode.

While EvoCF introduces additional modules beyond LLaMAR, it reduces planning steps per episode ($21.87 \rightarrow 18.69$), helping amortize the cost across fewer, higher-quality decisions. In other words, while EvoCF introduces slightly higher per-step latency and token cost, it enables faster and more reliable task completion overall, due to fewer total planning rounds and higher plan success.

### B.3 TABLE OF SYMBOLIC CONSTRAINTS INDUCED

Table 9 summarizes the symbolic constraints that were automatically induced based on failure feedback across different collaboration tasks. These results comprehensively cover the six key metrics introduced in Section 3.2, which fall under the two major categories of *Task-Dependency Constraints* and *Multi-Agent Coordination Constraints.*

| Task | Symbolic Constraints |
|---|---|
| ActionValidity | BlockedPath(?agent, ?destination) $\rightarrow$ ExploreOrReport(?agent); 
 ¬Nearby(?agent, ?object) $\rightarrow$ ChangeViewAngle(?agent, ?angle); 
 CheckLightStatus(?agent) $\rightarrow$ CorrectCommand(?agent); 
 Clean(?agent, ?object) $\rightarrow$ Holding(?agent, ?object); |
| ObjectReachability | AccessObject(?agent, ?object) $\rightarrow$ ClearPath(?agent, ?object); 
 AccessObject(?agent, ?object) $\rightarrow$ Open(?container); 
 CleanObject(?agent, ?object) $\rightarrow$ AtLocation(?agent, ?object); 
 CleanObject(?agent, ?object) $\rightarrow$ CloseTo(?agent, ?object); |
| SpatialFeasibility | Access(?agent, ?location) $\rightarrow$ ¬Blocked(?agent, ?location); 
 AdjustPosition(?agent) $\rightarrow$ ClearPath(?agent, ?destination); 
 AdjustPosition(?agent, ?destination) $\rightarrow$ Facing(?agent, ?destination); 
 Align(?agent, ?object) $\rightarrow$ ClearPath(?agent, ?object); 
 Aligned(?agent, ?object, ?target) $\wedge$ CloseEnough(?agent, ?target) 
 $\rightarrow$ Interact(?agent, ?object); 
 Aligned(?agent, ?target) $\rightarrow$ Move(?agent, ?direction); |
| TemporalConsistency | Explore(?agent) $\rightarrow$ Rotate(?agent) $\wedge$ Move(?agent); 
 Explore(?agent, ?area) $\rightarrow$ Open(?agent, ?drawer); 
 Interacting(?agent, ?object) $\wedge$ Interacting(?agent, ?object) $\rightarrow$ 
 Interference(?agent, ?agent); 
 Open(?agent, ?object) $\rightarrow$ Before(PutObject(?agent, ?object)); 
 OpenObject(?agent, ?drawer) $\rightarrow$ Occupied(?drawer); |
| LoadBalance | Assist(?agent1, ?agent2) $\rightarrow$ Idle(?agent1) $\wedge$ NeedAssistance(?agent2); 
 BetterPosition(?agent1, ?agent2, ?object) $\rightarrow$ 
 Assist(?agent1, ?agent2); 
 Closer(?agent1, ?object) $\wedge$ CanOpen(?agent1, ?object) $\rightarrow$ 
 Open(?agent1, ?object); 
 Closer(?agent1, ?location) $\wedge$ Blocked(?agent2, ?location) $\rightarrow$ 
 Help(?agent1, ?agent2); |
| GoalAlignment | Assist(?agent1, ?agent2) $\rightarrow$ ClearSpace(?agent1, ?agent2); 
 Assist(?agent1, ?agent2, ?task) $\rightarrow$ GoalAlignment(?agent1, ?task); 
 Assist(?agent1, ?agent2, ?task) $\rightarrow$ Near(?agent1, ?task); 
 Assist(?agent1, ?agent2, Locate(?object)) $\rightarrow$ Near(?agent1, ?object); |

Table 9: Inducted symbolic constraints across MAP-THOR tasks.

### USE OF LARGE LANGUAGE MODELS

We used large language models solely as auxiliary tools to assist in the language polishing and stylistic refinement of the paper draft. The research ideas, methodology, experiments, and analysis were entirely conceived and conducted by the authors. The LLM did not contribute to the design of the framework, the experimental results, or the interpretation of findings. All scientific content remains the responsibility of the authors.

