# OpenReview forum: "EvoCF: Multi-agent Collaboration with Memory-guided Evolutionary Counterfactual Planning"
_ICLR.cc/2026/Conference — Submitted to ICLR 2026_

### Official Review · Reviewer_WJsZ · 2025-10-22

**Soundness:** 2
**Presentation:** 3
**Contribution:** 2
**Rating:** 2
**Confidence:** 5

**Summary:**

The paper proposes a three part pipeline for multi-agent embodied planning: symbolic constraints induced from failure experiences, counterfactual plan generation using rule guided mutations, and a memory guided evaluator. The structure is clear and the MAP-THOR results are promising.

**However, ablation coverage is incomplete.** The paper does not isolate the contribution of each component, does not study operator sensitivity, and does not report token or latency cost.

**Positioning is also incomplete.** Closely related work in the Multi-LLM Agent Collaborative Intelligence literature is not cited or compared. This includes the MACI book (first edition March 2023, ACM Books release slated for end of 2025), SocraSynth and CRIT (2023) for counterfactual probing and rubric based judging, EVINCE (preprint) for modulation of contentiousness as a control dial in addition to information retrieval, ALAS (preprint) for disruption aware planning and compensation policies, SagaLLM: Context Management, Validation, and Transaction Guarantees for Multi-Agent LLM Planning (PVLDB 2025) for persistent execution memory with validation, rollback, and compensation, and A Checks-and-Balances Framework for Context-Aware Ethical AI Alignment (ICML 2025) for counterfactual reasoning in ethics alignment.

Please add a clear comparison and a novelty paragraph that states what is new here beyond those items, and provide module-level ablations, scaling beyond two agents, transfer to larger layouts, and basic cost reporting. See details in the rest of this review.

**Provisional rating pending rebuttal.**

**Strengths:**

* Clear modularization: rule induction from failures, rule guided counterfactual generator, memory guided evaluator.

* Concreteness: explicit mutation operators and an interpretable rule library that others can reuse.

* Positive results on MAP-THOR with a clean experimental setup.

**Weaknesses:**

**1. Missing citations and positioning:** The paper does not cite or compare against key MACI references that cover the same pillars the paper relies on. Please add and discuss:

* MACI book, first edition March 2023, ACM Books release slated for 2025.

* SocraSynth and CRIT (2023): counterfactual probing and rubric based judging.

* EVINCE (preprint): modulation of contentiousness to balance exploration and convergence in addition to information retrieval.

* ALAS (preprint): disruption aware planning with explicit compensation policies.

* SagaLLM: Context Management, Validation, and Transaction Guarantees for Multi-Agent LLM Planning (PVLDB 2025): persistent execution memory with validation, rollback, and compensation.

* A Checks-and-Balances Framework for Context-Aware Ethical AI Alignment (ICML 2025): counterfactual reasoning applied to ethics alignment.

**2. Ablation coverage is incomplete:** no direct isolation of the symbolic constraint inductor; no sensitivity to rule-library size or quality; no operator-level analysis (swap, insert, delete, replace); no study of memory retrieval depth or neighborhood size; no comparison of evaluator variants beyond a single proxy; no discussion of failure modes tied to each module.

**3. Limited scope:** results focus on two agents and AI2-THOR layouts. Scaling to more agents, larger environments, or real robots is not demonstrated. Transfer of induced rules across task families is not tested.

**4. No cost reporting:** token counts and wall-clock latency for generation, mutation search, and evaluation are missing.

**5. Limitations:** there is no dedicated limitations section that addresses generalization beyond hand-designed operators, robustness under perception noise or tool errors, and brittleness of induced rules.

**Provisional recommendation** pending a clear novelty statement relative to MACI and the requested ablations, scaling, transfer, cost, and limitations. I will revisit after rebuttal.  **The MACI book is in the public domain.**

**Questions:**

**Encouragement to authors**

The core pipeline is clear and the MAP THOR results are promising. Although the current ablations are limited, a stronger related work section can materially improve the paper. Explicitly citing and positioning against the MACI literature will help readers see this work as a focused engineering integration rather than uncredited reinvention. It also lets you highlight what is uniquely yours in this setting, for example the specific mutation operators, the failure induced rule library, and the MAP THOR integration.

1. Cite and compare against the MACI line listed above, and state precisely what is new here?

2. Provide module-level ablations: rule induction on vs off with the same generator and evaluator; operator sensitivity; memory retrieval depth; evaluator variants with and without retrieval.

3. Report token usage and latency for each phase (your competing papers did)

4. Evaluate scaling to more agents and larger environments, and test cross-domain transfer of induced rules.

5. Add a dedicated limitations section that covers generalization, robustness, and failure modes.

---

> ### Author Response · Authors · 2025-11-21
>
> Thank you sincerely for your thoughtful review and generous feedback. We have carefully addressed each of your comments, including **adding missing references** and discussing related literature in more depth. In addition, we have conducted new experiments and analyses to strengthen our empirical support. All revisions have been incorporated into the updated PDF and highlighted for your convenience (Please note that all line numbers mentioned correspond to the revised PDF).
>
> ---
>
> **Weakness1 & Question 1:**
> >*Missing citations and positioning.*
>
> **Response**:
>
> We thank the reviewer for highlighting the MACI literature and its contributions to multi-LLM agent collaborative intelligence. MACI is related to EvoCF in the sense that both pursue multi-agent LLM collaboration intelligence, but with distinct goals: MACI focuses on achieving human-like general intelligence, whereas EvoCF aims to discover better collaborative strategies in embodied environments.
>
> We've **added all the referenced works to the work with appropriate discussions** (**`New content highlighted; see Related Work section, Line 130-142)`**： Technically, CRIT[1] and Checks-and-Balances[2] apply counterfactuals to language QA and ethics, while EvoCF uses them to generate alternative executable plans. SagaLLM[3] and ALAS[4] address multi-agent LLM planning through transactional memory and real-time validation; EvoCF improves it via counterfactual plan construction to discover better joint solutions.EVINCE[5] and SocraSynth[6] modulate dialogue via retrieval; EvoCF retrieves past outcomes to evaluate candidate plans.
>
> We sincerely thank the reviewer for prompting a sharper articulation of our contributions.
>
> ---
>
> **Weakness2 & Question 2:**
> >*Ablation coverage is incomplete: provide module-level ablations: rule induction on vs off with the same generator and evaluator; operator sensitivity; memory retrieval depth; evaluator variants with and without retrieval.*
>
> **Response**:
>
> **1. Rule Induction Ablation:** We added a new ablation disabling the symbolic rule inductor while keeping the generator and evaluator fixed. As shown in **`Table 5 (New content highlighted; see Line 436)`**, ablating rule induction removes structural guidance for mutation and deprives the evaluator of reliable constraints, resulting in less targeted generation and weaker plan selection.
>
> | Table5| SR↑ | TR↑ |C↑| B↑| L↓ |
> | -  | -  | -  |- | - | - |
> | EvoCF(w/o Rule Induction)  | 0.75 | 0.93 | 0.98  | 0.87  | 19.62 |
> | EvoCF |  0.84  |   0.95   |     0.99      |   0.89    |  18.69  |
>
> **2. Operator Sensitivity:** We recorded `Occurrence` and `Adoption` of each mutation operator. As shown in **`Table 2 (New content highlighted; see Line 378-383)`**. **Occurrence Frequency** refers to the percentage of generated counterfactual plans that include each mutation operator; **Adoption Frequency** refers to the proportion of counterfactual plans  counterfactual plans produced by each operator are selected for execution, where **57\%** triggers original plan.
>
> | Table2    | Swap | Insert |Delete| Replace|
> | -  |-  | -  |- | - |
> | Occurrence Frequency  | 12% | 39% | 22%  | 27%  |
> | Adoption Frequency |  8%  |   17%   |     8%      |   10%    |
>
> Specifically, `Insert` is most impactful, enabling fine-grained corrections (e.g., Rotate/Move). `ReplaceObject` and `Delete` help resolve collisions or object mismatches. `SwapAgent` contributes less, as initial role allocation is often valid. The results show that the counterfactual planning replaces original plan with a noticeable high chance of **43\%**, highlighting the crucialness of counterfactual mutation in improving multi-agent planning robustness and flexibility.
>
> **3. Memory Retrieval Depth:**
> We added a new experiment varying Top-K. Results show that performance improves up to K = 5 and plateaus thereafter, suggesting that additional trajectories may introduce redundancy or noise **`(New content highlighted; see Table4 Lines 419-424)`**.
>
> | Table4    | K=1 |K=3| K=5(default)|K=10 |
> | -  |- |-  |- | - |
> | Success Rate (SR↑)| 0.77 | 0.81 | 0.84  | 0.82  |
>
>
> **4. Evaluator variants with and without retrieval:**
>
> We would like to clarify that our existing variant  `EvoCF (w/ Heuristic Evaluator)`**`(Previously included, Table5, line435)`** corresponds to the non-retrieval evaluator. Performance drops from 0.84 → 0.72 (SR), confirming the benefit of episodic memory.
>
> ---

---

> > ### Author Response · Authors · 2025-11-21
> >
> > **Weakness4 & Question 3:**
> > >*Report token usage and latency for each phase.*
> >
> > **Response**:
> >
> > We added per-module latency and token cost in  **`AppendixB.2 (New content highlighted; see Table7 & Table8, Lines839-860)`**. Given that EvoCF employs LLaMAR as its LLM planner backbone, its additional counterfactual planning modules (rule induction, plan generation, and evaluation) introduce only a reasonable overhead in both latency and token usage. As shown in the Table below, EvoCF reduces planning steps per episode (21.87 → 18.69), helping amortize the cost across fewer, higher-quality decisions.
> >
> > | Table7 (latency)   | LLaMAR-Moudles| Constraint Inductor|Counterfactual Plan Generator | Evaluator| Step | Toal |
> > | -  | -  | - |- |-  |-  |- |
> > | LLaMAR             | 7.61 | - | -  | -  | 21.87 | 166.43 |
> > | EvoCF(ours) | 7.61  |  1.46   |    2.17    |   2.31      |  18.69  |   269.76|
> >
> > | Table8 (token)   | LLaMAR-Moudles| Constraint Inductor|Counterfactual Plan Generator | Evaluator| Step | Toal |
> > | -  | -  | - |- |-  |-  |- |
> > | LLaMAR             | 1580|- |- | - | 21.87 | 35.6K |
> > | EvoCF(ours) |   1580   |    245    |  427   |  512  | 18.69      |   47.8K    |
> >
> > Beyond efficiency, EvoCF yields two key gains that dominate costs in multi-agent and real-world deployments:
> >
> > **(1) Coordination stability:** higher balance and tighter synchronization reduce congestion and communication rounds, avoiding deadlocks/rollbacks.
> > **(2) Fewer within-round failures:** the evaluator prunes failure-prone plans before execution, cutting retries and safety interventions.
> >
> > These effects make EvoCF not just faster in expectation, but also more reliable and economical when execution and recovery are expensive.
> >
> > ---
> >
> > **Weakness3 & Question 4:**
> > >*Limited scope.*
> >
> >
> > **Response**:
> >
> > **(a) Scaling to more agents:**
> >
> > We respectfully highlight that the results on scaling to more agents has already been reported in the  **`Table 3 (Previously included, Line393-400)`**. EvoCF performs consistently across 2, 3, and 4-agents settings: SR improves to 0.87 with 3 agents and remains high (0.82) with 4. The drop in balance and slight rise in planning steps at 4 agents reflect coordination complexity under the fixed step cap (L=30), not computational limits.
> >
> > | # Agents| SR↑ | TR↑ |C↑| B↑| L↓ |
> > | -  | -  | -  |- | - | - |
> > | 2 |  0.84  |   0.95   |     0.99      |   0.89    |  18.69  |
> > | 3 |  0.87  |   0.96   |     0.99      |   0.81    |  17.36  |
> > | 4 |  0.82  |   0.93   |     0.99      |   0.74    |  19.51  |
> >
> > **(b) Scaling to larger environments:**
> > Each MAP-THOR task spans 3–5 layouts of varying size and complexity, naturally testing environment scaling. To our knowledge, MAP-THOR remains the most relevant and largest available benchmark for multi-agent interaction and collaboration.
> >
> > **(c ) Rule transferability across tasks:**
> > We would also like to clarify that this experiment has already been reported in **`Appendix B.1 (Previously included, Table 6, line826-835)`**.
> >
> > To test the transferability of symbolic constraints, we conduct a minimal intervention study: rules induced from a source task `(put food in fridge)` are directly injected into planning for two unrelated tasks `(put pots on stove, open all drawers)`, without invoking counterfactual generation or inducing new rules.
> >
> > As shown in Table below, performance improves across all metrics. Success rate increases from 0.60→0.80 and 0.40→0.60, respectively, with gains in transport rate, balance, and efficiency, confirming that EvoCF’s symbolic rules generalize across tasks with different semantics and layouts.
> >
> > | Target Task | Setting | SR↑ | TR↑ |C↑| B↑| L↓ |
> > | -  | -  | -  |- | - | - | - |
> > | Put the pots and pans on the stove |  Planner Only  |   0.60   |     0.88      |0.95  | 0.80    |  20.13  |
> > | Put the pots and pans on the stove |  w/ Transferred Constraints |   0.80   |     0.93      |   0.97    |  0.82  | 18.55 |
> >
> > | Target Task | Setting | SR↑ | TR↑ |C↑| B↑| L↓ |
> > | -  | -  | -  |- | - | - | - |
> > |Open all drawers|  Planner Only  |   0.40   |     0.75      |0.94  | 0.75   |  22.41  |
> > | Open all drawers |  w/ Transferred Constraints |   0.60   |     0.88      |   0.94    |  0.79  | 20.95 |

---

> ### Author Response · Authors · 2025-11-21
>
> **Weakness5 & Question 5:**
> >*Add limitations.*
>
> **Response**:
>
> Thank you for your comments. The issues you raised were briefly discussed at the end of our experimental section, but we acknowledge this may not have been sufficiently visible. To address this, we have added a dedicated **`Limitations section (Lines 516–525)`**, discussing the generalization challenges of mutation operators, the scope of induced constraints, and the roubtness of memory-based evaluation. The manuscript has been updated accordingly.
>
> ---
>
> [1]E. Chang. Multi-LLM Agent Collaborative Intelligence: The Path to Agi. ACM Books. Association for Computing Machinery, 2025a.
>
> [2]Chang E Y. Prompting large language models with the socratic method[C]//2023 IEEE 13th annual computing and communication workshop and conference (CCWC). IEEE, 2023: 0351-0360.
>
> [3]Chang E Y. A Checks-and-Balances Framework for Context-Aware Ethical AI Alignment[J]. arXiv preprint arXiv:2502.00136, 2025.
>
> [4]Chang E Y. SagaLLM: Context Management, Validation, and Transaction Guarantees for Multi-Agent LLM Planning[J]. Proc. VLDB Endow.
>
> [5]Geng L, Chang E Y. ALAS: Transactional and Dynamic Multi-Agent LLM Planning[J]. arXiv preprint arXiv:2511.03094, 2025.
>
> [6]Chang E Y. Evince: Optimizing adversarial llm dialogues via conditional statistics and information theory, 2024[J]. URL https://arxiv. org/abs/2408, 14575.
>
> [7]Chang E Y. SocraSynth: Multi-LLM reasoning with conditional statistics[J]. arXiv preprint arXiv:2402.06634, 2024.
>
> ---
>
> **We sincerely appreciate Reviewer WJsZ’s careful review and insightful feedback. Reviewer WJsZ's comments reflect deep expertise in multi-agent intelligence, and we feel honored to engage in this dialogue.  You not only provided a comprehensive list of relevant references across the seminal MACI book on multi-agent LLM intelligence, but also offered thoughtful suggestions on experimental design and evaluation, which directly helped us improve the clarity, completeness, and rigor of our work, for which we are truly grateful.**
>
> **Following your suggestions, we have added the requested ablations and analyses: `Operator Sensitivity (Table 2)`, `Scaling to More Agents (Table 3)`, `Memory Retrieval Depth Sensitivity (Table 4)`, `Rule Induction Ablation (Table 5)`,  `Rule Transferability Case Study (Table 6)`, and `Latency and Token Cost Report (Tables 7 & 8)`, for your kind review. We hope the additional materials and revisions provide a useful and meaningful response to your comments, and we would be delighted to engage further on any outstanding questions.**
>
> ---

---

> > ### Comment · Reviewer_WJsZ · 2025-11-26
> >
> > Thank the authors' responses.  Since ICLR permits the authors to upload an update by 12/2, if you could provide such, I can re-evaluate my rating.

---

> > > ### Author Response · Authors · 2025-11-28
> > > **Warm Thanks and Update: Revision PDF Now Live on OpenReview**
> > >
> > > Thank you very much for your thoughtful follow-up. **We warmly note that we’ve uploaded a new revised PDF to OpenReview, with all newly added content clearly highlighted for your convenience.**
> > >
> > > Following your helpful suggestions, we’ve added a comparison with the MACI and its associated works in the **Related Work section**, as well as new experiments covering **operator sensitivity**, **scaling to more agents**, **memory retrieval depth sensitivity**, **rule induction ablation**, **rule transferability**, and **latency & token cost reporting**.
> > >
> > > We sincerely appreciate your willingness to offer a reevaluation of the paper. Your constructive feedback directly helped us improve the clarity, completeness, and rigor of the paper, and we are truly grateful for the time and care you’ve dedicated throughout the process.

---

### Official Review · Reviewer_1hYG · 2025-10-28

**Soundness:** 3
**Presentation:** 3
**Contribution:** 2
**Rating:** 4
**Confidence:** 3

**Summary:**

This paper introduces EvoCF, a new framework for multi-agent collaboration planning that addresses the limitations of current LLM-based planners in complex environments. EvoCF leverages a structured memory of past experiences—particularly failures—to induce symbolic constraints that represent critical inter-agent dependencies and action feasibility. Using evolutionary algorithms, it systematically generates and explores alternative joint plans through counterfactual mutations, guided by these learned rules. A memory-driven evaluator then scores each candidate plan for robustness, efficiency, and coordination. Experimental results on simulation benchmarks show that EvoCF consistently produces more reliable, executable, and adaptable plans than existing state-of-the-art approaches.

**Strengths:**

The authors present a creative integration of structured symbolic memory, rule induction from failure experiences, and evolutionary counterfactual planning for multi-agent collaboration. The combination distinguishes it from prior work. EvoCF is original in its systematic exploration of alternative joint plans using learned constraints, moving beyond conventional single-shot or heuristic LLM-based planning. Introducing explicit counterfactual reasoning and evolutionary search into the LLM multi-agent planning paradigm represents a fresh perspective. Overall I think the paper is highly original in its problem framing and solution design, maintains methodological quality and sound experimental validation, and communicates its ideas clearly.

**Weaknesses:**

- The symbolic rule induction process appears fundamentally reliant on the presence and diversity of failure cases in structured memory, yet the paper does not address how EvoCF copes with sparse, unbalanced, or noisy episodic data. This should be a common scenario as environments grow complex or as agents encounter novel tasks.
- The constraint induction mechanism, while formally defined, seems constrained to a set of relatively simple precondition and coordination rules, potentially missing high-level, non-obvious dependencies or temporally extended causal relations essential for advanced multi-agent coordination. Mutation operators in the evolutionary plan generator are limited to a small, discrete set of modifications (e.g., SwapAgent, InsertAction), which may not be sufficiently expressive or efficient to navigate the combinatorial explosion of plausible joint plans as agent and action space scales, or to handle rich, long-horizon tasks.
- I'm not sure but the reliance on retrieval-augmented plan evaluation side-steps explicit world-modeling may fail catastrophically when retrieved experiences do not closely match the ongoing scenario. Critically, the paper does not report the computational efficiency of evolutionary plan search, nor does it clarify whether the constraint evaluation and plan ranking can be kept tractable in non-trivial settings.

Some of these points may not reflect actual weaknesses, as certain technical details were not explicitly provided in the paper. I would be glad to reconsider my score if the authors can offer further clarification or additional evidence addressing these concerns.

**Questions:**

- Can you provide more detail about the complexity of the simulation environments used in your benchmarks? How many agents, objects, and concurrent dependencies are typical, and how do they compare to real-world multi-agent scenarios?
- How sensitive is EvoCF to the size, diversity, and quality of the episodic memory? What happens if failure cases are under-represented or the memory buffer is sparse or noisy?
- What are the computational costs (in terms of time and resources) of EvoCF's evolutionary plan search and retrieval-augmented evaluation, especially as the number of agents or the complexity of tasks increases?
- Can you share some concrete examples (or just some logs) of induced symbolic rules from real experiments? I'm curious how expressive or general are these rules compared to hand-crafted domain knowledge.

---

> ### Author Response · Authors · 2025-11-21
>
> Thank you for your thoughtful review and encouraging assessment. Below, we respond to all your comments with detailed clarifications and  we have conducted new experiments and analyses to strengthen our empirical support. The newly added revisions have been updated in the PDF and highlighted for your reference. (Please note that all line numbers mentioned correspond to the revised PDF)
>
> ---
>
> **Weakness 1:**
> >*The symbolic rule induction process appears fundamentally reliant on the presence and diversity of failure cases in structured memory, yet the paper does not address how EvoCF copes with sparse, unbalanced, or noisy episodic data. This should be a common scenario as environments grow complex or as agents encounter novel tasks.*
>
>
> **Response**:
> We appreciate the opportunity to clarify how EvoCF handles such situations.
>
> **1. Dependence on failure data quality and diversity**
> While EvoCF induces rules solely from failure episodes, it analyzes structured memory within the same trajectory, including preconditions, effects, and context, not just the failed transition itself, to infer causal failure patterns beyond surface-level signals.
>
> **2. Noise filtering and sparsity handling.**
> Each rule is annotated with a confidence score and support count, allowing EvoCF to prioritize reliable constraints and filter noise. When memory is sparse or unreliable, the evaluator falls back to symbolic checks, favoring conservative plans over risky ones.
>
> ---
>
> **Weakness 2:**
>
> >*2.1 The constraint induction mechanism seems constrained to a set of relatively simple precondition and coordination rules, potentially missing high-level, non-obvious dependencies or temporally extended causal relations essential for advanced multi-agent coordination.*
>
>
> >*2.2 Mutation operators in the evolutionary plan generator are limited to a small, discrete set of modifications , which may not be sufficiently expressive or efficient to navigate the combinatorial explosion of plausible joint plans as agent and action space scales, or to handle rich, long-horizon tasks.*
>
> **Response**:
>
> Thank you for highlighting this important point.
>
> **2.1 Constraint expressiveness:**
> As shown in **`Appendix B.3(Table9, Line870-905)`**, EvoCF captures six categories of constraints, including temporal, spatial, and goal alignment rules..., providing broad structural coverage.  EvoCF induces rules that go beyond shallow action-level constraints. For example,
>
> >**`Open(?agent, ?object) →
> Before(PutObject(?agent, ?object))`** encodes temporally extended causal relations.
>
> >**`Assist(?agnet1, ?agent2) → ClearSpace(?agent1, ?agent2)`** reflects high-level coordination logic.
>
> These demonstrate EvoCF's ability to abstract structured, multi-agent dependencies across time and roles.
>
> **2.2 Mutation operator coverage:**
>
> We propose four mutation operators, **`{SwapAgent, InsertAction, ReplaceObject, DeleteAction}`**, which are sufficient to cover all symbolic variations of plans at the action-level in the MAP-THOR domain.
>
> In MAP-THOR, each action is structured as **`Act(agent, object) `**, where the action space is discrete and verb-object grounded. Within this representation, our operator set allows the LLM to explore and recombine plan elements flexibly, **while remaining sufficiently expressive and efficient to navigate the combinatorial space of joint plans as the number of agents and planning horizon increases.**
>
>
> We further support this with usage statistics in **`Table 2 (New Line 378–383)`**, which shows the occurrence and adoption percentage of counterfactual plans based on each operator. Notably, InsertAction accounts for 39% of mutations and 17% of selected plans, enabling nuanced refinements like spatial adjustment. The consistent contribution of all operators indicates their utility across diverse plan-editing scenarios.
>
> | Table2    | Swap | Insert |Delete| Replace|
> | -  |-  | -  |- | - |
> | Occurrence Frequency  | 12% | 39% | 22%  | 27%  |
> | Adoption Frequency |  8%  |   17%   |     8%      |   10%    |
>
> ---

---

> > ### Author Response · Authors · 2025-11-21
> >
> > **Weakness 3:**
> > >*Reliability & tractability of retrieval-based evaluation.*
> >
> >
> > **Response**:
> >
> > Thank you for raising this important point. We fully agree that relying solely on surface-level retrieval can lead to brittleness, especially in out-of-distribution or novel scenarios. However, we would like to clarify that **EvoCF's retrieval-augmented evaluator is not a nearest-neighbor matching mechanism, but rather a structured, constraint-guided reasoning module.**
> >
> > Specifically, the evaluator aggregates relevant prior outcomes and symbolic rules from memory to inform a **multi-objective plan ranking** function **`(Eq. 4， line315)`**. The symbolic constraints act as a primary filter, and retrieved outcomes provide contextual support for evaluating feasibility, coordination, and goal progress. Importantly, our compositional query design retrieves evidence at varying levels of abstraction, allowing the system to generalize beyond exact matches.
> >
> > Furthermore, **our ablation study demonstrates that EvoCF’s evaluator consistently selects higher-quality plans even when counterfactuals are noisy or the mutation space is diverse.** In practice, when retrieval confidence is low, the system tends to prefer conservative, lower-risk candidates—thus avoiding catastrophic failure modes.
> >
> > That said, we appreciate the reviewer’s concern and fully agree that future extensions incorporating learned world models or simulation-based validation would further improve robustness in novel scenarios. Your feedback has been instrumental in helping us think more deeply about this limitation and its mitigation.
> >
> > ---
> >
> > **Question 1:**
> > >*Provide more detail about the the environments.*
> >
> >
> > **Response**:
> >
> > Our experiments use the **MAP-THOR benchmark**[1], features  multi-embodied agents operating under partial observability in photorealistic **AI2-THOR**[2] scenes (kitchen, bedroom, living room, bathroom). Each floor plan contains 25–45 manipulable objects and dense spatial layouts that induce realistic navigation interference when >2 agents are present. Tasks decompose into interdependent subtasks (e.g., one agent opens a drawer, another stores items), creating concurrent temporal and spatial dependencies. This setup reflects the coordination, congestion, and uncertainty typical of real-world multi-agent environments.
> >
> > ---
> >
> > **Question 2:**
> >
> > >*How sensitive is EvoCF to the size, diversity, and quality of the episodic memory? What happens if failure cases are under-represented or the memory buffer is sparse or noisy?*
> >
> > **Response**:
> >
> > We thank the reviewer for raising this important point. We assess sensitivity to memory **quality** and **size** via two experiments:
> >
> > **Memory sensitivity analysis.**
> >
> > **1.Retrieval depth `(New content highlighted; see Table 4, Lines419-424)`**: Increasing the number of retrieved trajectories from 1 to 5 improves success rate (0.77 → 0.84), suggesting that the evaluator benefits from diverse past experiences. Performance drops slightly at K=10, indicating diminishing returns or noise. This shows EvoCF is robust to moderate memory variation and uses confidence-weighted evidence to filter noise.
> >
> > | Table4    | K=1 |K=3| K=5(default)|K=10 |
> > | -  |- |-  |- | - |
> > | Success Rate (SR↑)| 0.77 | 0.81 | 0.84  | 0.82  |
> >
> >
> > **2.Rule transferability `(Previously included, see Appendix B.1, Table 6, lines826-835)`**: Constraints induced from a single unrelated task significantly improve performance on new tasks, without invoking new memory or retrieval. This demonstrates that EvoCF’s symbolic constraints are generalizable and effective even under sparse or missing episodic data.
> >
> > | Target Task | Setting | SR↑ | TR↑ |C↑| B↑| L↓ |
> > | -  | -  | -  |- | - | - | - |
> > | Put the pots and pans on the stove |  Planner Only  |   0.60   |     0.88      |0.95  | 0.80    |  20.13  |
> > | Put the pots and pans on the stove |  w/ Transferred Constraints |   0.80   |     0.93      |   0.97    |  0.82  | 18.55 |
> >
> > | Target Task | Setting | SR↑ | TR↑ |C↑| B↑| L↓ |
> > | -  | -  | -  |- | - | - | - |
> > |Open all drawers|  Planner Only  |   0.40   |     0.75      |0.94  | 0.75   |  22.41  |
> > | Open all drawers |  w/ Transferred Constraints |   0.60   |     0.88      |   0.94    |  0.79  | 20.95 |
> >
> > These results jointly support that EvoCF remains effective under limited or imperfect memory, and we will clarify this further in the final version.
> >
> > ---

---

> > > ### Author Response · Authors · 2025-11-21
> > >
> > > **Question 3:**
> > > >*Computational cost of EvoCF, especially as agent count or task complexity increases?*
> > >
> > >
> > > **Response**:
> > > We added per-module latency and token cost in  **`AppendixB.2 (New content highlighted; see Table7 & Table8, Lines839-860)`**. Given that EvoCF employs LLaMAR as its LLM planner backbone, its additional counterfactual planning modules (rule induction, plan generation, and evaluation) introduce only a reasonable overhead in both latency and token usage. As shown in the Table below, EvoCF reduces planning steps per episode (21.87 → 18.69), helping amortize the cost across fewer, higher-quality decisions.
> > >
> > > | Table7 (latency)   | LLaMAR-Moudles| Constraint Inductor|Counterfactual Plan Generator | Evaluator| Step | Toal |
> > > | -  | -  | - |- |-  |-  |- |
> > > | LLaMAR             | 7.61 | - | -  | -  | 21.87 | 166.43 |
> > > | EvoCF(ours) | 7.61  |  1.46   |    2.17    |   2.31      |  18.69  |   269.76|
> > >
> > > | Table8 (token)   | LLaMAR-Moudles| Constraint Inductor|Counterfactual Plan Generator | Evaluator| Step | Toal |
> > > | -  | -  | - |- |-  |-  |- |
> > > | LLaMAR             | 1580|- |- | - | 21.87 | 35.6K |
> > > | EvoCF(ours) |   1580   |    245    |  427   |  512  | 18.69      |   47.8K    |
> > >
> > > Beyond efficiency, EvoCF yields two key gains that dominate costs in multi-agent and real-world deployments:
> > >
> > > **(1) Coordination stability:** higher balance and tighter synchronization reduce congestion and communication rounds, avoiding deadlocks/rollbacks.
> > > **(2) Fewer within-round failures:** the evaluator prunes failure-prone plans before execution, cutting retries and safety interventions.
> > >
> > > These effects make EvoCF not just faster in expectation, but also more reliable and economical when execution and recovery are expensive.
> > >
> > > ---
> > >
> > > **Question 4:**
> > > >*Share some examples of induced symbolic rules.*
> > >
> > >
> > > **Response**:
> > >
> > > We provide concrete examples of induced symbolic rules in **`Appendix B.3 (Table 9, lines 870–905)`**, spanning six categories. Here are three real examples:
> > >
> > > >**Temporal Consistency**
> > > **`Open(?agent, ?object) → Before(PutObject(?agent, ?object))`**
> > >
> > > Captures causal ordering in manipulation.
> > >
> > > >**Spatial Feasibility**
> > > **`AdjustPosition(?agent, ?destination) → Facing(?agent, ?destination)`**
> > >
> > >  Ensures spatial alignment before interaction.
> > >
> > > These rules are induced automatically from episodic traces, abstracted over variable bindings, and reused across diverse tasks, unlike hand-crafted rules, they reflect emergent, data-driven patterns.
> > >
> > > ---
> > >
> > > We sincerely thank Reviewer 1hYG for your constructive and detailed feedback. Your suggestions have been invaluable in guiding our revisions, and we would be delighted to engage further on any outstanding questions.
> > >
> > > ---
> > >
> > > [1]Nayak S, Orozco A M, Ten Have M, et al. MAP-THOR: Benchmarking Long-Horizon Multi-Agent Planning Frameworks in Partially Observable Environments[C]//Multi-modal Foundation Model meets Embodied AI Workshop@ ICML2024.
> > >
> > > [2]Kolve E, Mottaghi R, Han W, et al. Ai2-thor: An interactive 3d environment for visual ai[J]. arXiv preprint arXiv:1712.05474, 2017.

---

### Official Review · Reviewer_HnHQ · 2025-11-01

**Soundness:** 2
**Presentation:** 3
**Contribution:** 3
**Rating:** 4
**Confidence:** 3

**Summary:**

This paper introduces EvoCF, a memory-guided framework for discovering improved multi-agent collaboration strategies through counterfactual plan generation and evaluation. Specifically, the framework includes three processes: (i) the Counterfactual Plan Generator, which introduces evolutionary operators to explore diverse constraint-guided alternatives; (ii) the Retrieval-Augmented Counterfactual Evaluator, which grounds these candidates in past outcomes and symbolic constraints to assess their viability; and (iii) the Symbolic Constraint Inductor, which distills failure patterns into reusable rules that accumulate in memory. Experiments on MAP-THOR reflect the effectiveness of EvoCF compared to various baselines.

**Strengths:**

The reliability of planning among multiple agents is a critical issue in real-world scenarios. This paper addresses this by leveraging historical experiences, constructing rule-based memory, employing retrieval mechanisms tailored to task execution, and dynamically updating memory based on new execution outcomes. This constitutes an effective paradigm of agent evolution. Experimental results also demonstrate a significant improvement of this approach compared to the baseline method.

**Weaknesses:**

1. Section 3 employs an excessive number of concepts and symbols, and some definitions are not sufficiently clear. The relationships between symbols are ambiguous, making it very challenging for readers. For instance, what is the difference between $\psi$ in line 201 and $\phi$ in line 236? What distinguishes $\mathcal{R}$ in line 219 from $\Psi$ in line 210? How does the concept of memory differ from that of a library in the paragraph corresponding to line 192? How is $\Psi^{\rm{rule}}(m)$ operated in line 257, and what does the embedding function $e(.)$ specifically entail? What does the term "identity information" refer to in line 250? I recommend that the author avoid arbitrary noun substitutions, use consistent terminology and symbols for the same entity, and provide a complete example illustrating each part of the method with corresponding names and symbols to enhance the readability of the article.

2. In line 058, it is mentioned that one of the motivations of this study is the increasing importance of reliable planning in multi-agent planning as the number of agents grows. However, it appears that the experiments in this study only involve 2-4 agents, which does not adequately demonstrate the scalability of the approach concerning the number of agents.

3. The method in this paper is only validated on one dataset. Would the approach still be effective in a different scenario? Additionally, the experiments in the paper only use one model. How would the approach perform with open-source models?

4. There are some typos in Figure 1: "Experience-Given" should be "Experience-Driven," and "Rlues" should be corrected to "Rules."

**Questions:**

In many real-world scenarios, the consequences of actions are often irreversible. In such cases, what is the meaning of reflecting on and refining historical experiences? How can we enhance reliability by improving the accuracy of multi-agents' first-time actions?

---

> ### Author Response · Authors · 2025-11-21
>
> We sincerely thank you for your careful review and thoughtful understanding of our work. Below, we provide detailed responses to all your comments and describe the corresponding revisions made. (Please note that all line numbers mentioned correspond to the revised PDF)
>
> ---
>
> **Weakness 1:**
> >*Section 3 employs an excessive number of concepts and symbols, and some definitions are not sufficiently clear. The relationships between symbols are ambiguous, making it very challenging for readers.*
>
>
> **Response**:
> The following clarifies the distinctions among key symbols and concepts:
>
> ```
> [Memory 𝓜]
> └── stores episodic transitions m = (oₜ, aₜ, {ψₖ})
>     └── ψₖ ∈ Ψ ← Ψ (line 220): annotation type space
>         → e.g., ψ_plan, ψ_eff, ψ_fail (line 212)
> ```
>
> - **Memory (𝓜)** is the structured episodic store of transition-level records and annotations;
> - **ψ (line 212)** refers to structured annotations in memory (e.g., `ψ_plan`, `ψ_fail`, `ψ_eff`) that encode subgoal IDs, observed effects, or failure signals from past episodes. **Ψ (line 220)** denotes the type space of structural annotations.
>
> ```
> [Constraint Inductor]
>   └── C_gen(m) → rule r = (ϕ, τ, μ)
>         └── ϕ: symbolic formula (e.g., precondition/effect)  ← ϕ (line 247)
>         └── τ: rule type {dep, coord}
>         └── μ: confidence/support
>         → collected into global rule set ℛ ← ℛ (line 230)
> ```
>
> - **𝓡 (line 230)** is the evolving **rule library**: a global set of symbolic constraints induced across all experiences.
> - **ϕ (line 247)** is the symbolic formula in a learned constraint rule (e.g., `OpenObject(?obj) → Facing(?obj)`), abstracted from failure-annotated ψ entries.
>
> ```
> [Retrieval Module]
>   └── Query: q = AGG((x₁, x₂, ...), ξ)
>         └── ξ: identity info (e.g., agent, task)  ← ξ (line 261)
>         └── e(·): embedding function (e.g., SBERT) ← e(·) (line 267)
>
>   └── Retrieve: Top-k similar m ∈ 𝓜 → Ψ_rule(m)  ← Ψ_rule(m) (line 267)
>         → yields relevant rules from memory
> ```
> - **Identity information (ξ) (line261)** in the query `q = AGG((x₁, x₂, ..., xₖ), ξ)` encodes contextual metadata such as agent ID, room type, or task index to improve retrieval specificity.
> - **Ψ_rule(m) (line 267)** denotes the subset of rules induced from a specific memory item `m`, typically via the generator `C_gen(m)`. It is used during retrieval to provide local symbolic context.
> - The **embedding function `e(·)(line 267)`** maps a query or memory item into a shared semantic space (we use a frozen Sentence-BERT encoder for this), enabling top‑k retrieval based on cosine similarity.
>
> We appreciate the reviewer’s suggestion and will ensure the final version is more accessible to readers unfamiliar with our internal terminology.
>
> ---
>
> **Weakness 2:**
> >*However, it appears that the experiments in this study only involve 2-4 agents, which does not adequately demonstrate the scalability of the approach concerning the number of agents.*
>
>
> **Response**:
> Thank you for pointing this out.
>
> **MAP-THOR[1] is an embodied multi-agent benchmark designed for collaborative tasks in household environments.** Our experiments follow its standard configuration, **where 2–4 agents are commonly used to reflect realistic agent density in constrained settings.**
>
> We further emphasize that **MAP-THOR focuses on household embodied collaboration,** where tasks such as cooking, cleaning, or organizing are naturally performed by a **small number of agents.**  In this setting, EvoCF already capture the core coordination challenges, **while larger teams often introduce redundancy and idleness rather than scaling complexity**.
>
> That said, EvoCF is designed with general multi-agent scalability in mind: its constraint-guided mutation, retrieval-based evaluation, and decentralized symbolic reasoning mechanisms are model-agnostic and modular. We believe they can generalize to other domains with larger teams (e.g., logistics, industrial collaboration), which we view as a promising direction for future work.
>
> ---

---

> > ### Author Response · Authors · 2025-11-21
> >
> > **Weakness 3.1:**
> > >*3.1 The method in this paper is only validated on one dataset. Would the approach still be effective in a different scenario?*
> >
> > **Response to 3.1**:
> > Although we evaluate EvoCF on a single benchmark, MAP-THOR itself encompasses a diverse set of **45 tasks** across **multiple home environments and difficulty levels**. Each task is further instantiated with **3–5 different scene** initializations across varied household layouts, object placements, and agent starting positions. These tasks span **four distinct categories** based on the ambiguity of instructions:
> >
> > `1. Explicit items, quantities, and targets (e.g., “Put bread, lettuce, and a tomato in the fridge”)`
> >
> > `2. Explicit items but implicit quantities (e.g., “Put all the apples in the fridge”)`
> >
> > `3. Implicit items, explicit target (e.g., “Put all groceries in the fridge”)`
> >
> > `4. Implicit items and targets (e.g., “Clear the floor by placing items appropriately”)`
> >
> > This coverage includes long-horizon planning, implicit grounding, and ambiguity resolution, offering rich variations in object types, goals, and language grounding, all within realistic, partially observable 3D environments.
> >
> > **Weakness 3.2:**
> >
> > >*3.2 Additionally, the experiments in the paper only use one model. How would the approach perform with open-source models?*
> >
> >
> > **Response to 3.2**:
> > EvoCF is model-agnostic: it functions as a symbolic, memory-guided planning framework layered over any instruction-following model. Although we use GPT-4o for consistent comparison with prior work, EvoCF can be readily paired with open-source LLMs that support plan generation and feedback parsing.
> >
> >
> > **Weakness 4:**
> > >*There are some typos in Figure 1: "Experience-Given" should be "Experience-Driven," and "Rlues" should be corrected to "Rules."*
> >
> >
> > **Response**:
> > Thank you for your careful reading. We have corrected these typos and have carefully reviewed the entire manuscript to fix any remaining issues.
> >
> > ---
> >
> > **Question 1:**
> > >*In many real-world scenarios, the consequences of actions are often irreversible. In such cases, what is the meaning of reflecting on and refining historical experiences? How can we enhance reliability by improving the accuracy of multi-agents' first-time actions?"*
> >
> >
> > **Response**:
> >
> > Thank you for raising this insightful and expert-level question, anticipating failure in one-shot, irreversible scenarios is indeed a critical challenge in real-world planning.
> >
> > EvoCF’s LLM-based planner composes multi-agent action plans grounded in current state observations and high-level subgoals. **For first-time actions**, the LLM leverages **general world knowledge** and **commonsense reasoning** to infer plausible action sequences, even under novel combinations of agents, objects, and environments.
> >
> > Beyond this, EvoCF **retrieves structurally similar past episodes from memory**, providing supporting context, including symbolic annotations, prior failures, and local MDP features, that can guide better decision-making in new, unseen situations.
> >
> > Furthermore, EvoCF's symbolic constraint mechanism enables **proactive identification of potentially unsafe or suboptimal plans and suggests safer alternatives**. When action failures do occur, these constraints help prune the search space, accelerating recovery and fast local exploration.
> >
> > Finally, the reliability of first-time actions can be further enhanced by **few-shot prompting**, allowing it to generalize from curated demonstrations before any failures occur.
> >
> > ---
> > We are grateful to Reviewer HnHQ for your rigorous and thoughtful assessment. Your critique helped clarify key aspects of our contribution, and we look forward to continuing the conversation should any remaining issues remain.
> >
> > ---
> > [1]Nayak S, Orozco A M, Ten Have M, et al. MAP-THOR: Benchmarking Long-Horizon Multi-Agent Planning Frameworks in Partially Observable Environments[C]//Multi-modal Foundation Model meets Embodied AI Workshop@ ICML2024.

---

### Official Review · Reviewer_pAHc · 2025-11-02

**Soundness:** 3
**Presentation:** 3
**Contribution:** 3
**Rating:** 6
**Confidence:** 4

**Summary:**

The authors presents EvoCF (Evolutionary Counterfactual Planning), a memory-guided framework designed to address the challenges of planning and coordination for multi-agent embodied systems in partially observable environments. The core contribution is a deliberative, iterative loop that moves beyond single-shot LLM plans by integrating three novel components:

1. Symbolic Constraint Induction: A structured memory records failure experiences and extracts reusable symbolic rules that encode multi-agent coordination requirements and single-agent feasibility constraints.
2. Evolutionary Counterfactual Plan Generator: This module systematically explores alternative joint action plans by applying mutation operators (e.g., SwapAgent, InsertAction) to a seed plan. Mutations are guided by the retrieved symbolic rules to ensure semantic consistency and task relevance.
3. Experience-Driven Evaluator: Candidate plans are ranked based on memory-retrieved outcomes and learned symbolic constraints, effectively acting as a world-model-free reasoning layer to anticipate consequences and preemptively avoid failure.


Experiments on the MAP-THOR benchmark show EvoCF achieves an $18\%$ higher success rate than the strong baseline LLaMAR.

**Strengths:**

The framework introduces a novel and robust method for improving multi-agent planning by integrating explicit symbolic rule induction from failure experiences into the LLM planning loop, ensuring that coordination constraints are reusable and transferable across tasks.

• The use of evolutionary counterfactual search is a strong methodological step, moving beyond the inherent limitations of conventional one-shot LLM planners by systematically generating diverse, constraint-guided plan alternatives.

• The paper demonstrates that the induced symbolic constraints encode structurally generalizable knowledge. A dedicated case study confirms that rules learned from one task significantly improve the success rate on structurally different, unseen target tasks.

• The ablation study clearly validates the importance of both core modules: removing either the rule-guided generator or the experience-driven evaluator leads to a notable drop in performance, confirming their synergistic contribution to overall robustness.

• EvoCF achieves significant empirical gains, outperforming state-of-the-art baselines like LLaMAR by $18\%$ in success rate and substantially improving metrics across transport rate, coverage, and coordination balance in multi-agent embodied scenarios.

**Weaknesses:**

- The Symbolic Constraint Induction process is not fully detailed as a black box. The generator that maps a failure-annotated transition to candidate rules relies on an implicit LLM reasoning step, making the induction quality and reliability difficult to assess or reproduce without knowing the underlying prompting or training.
- The mutation operators are explicitly stated as manually designed (e.g., `SwapAgent`, `ReplaceObject`), and the ablation study confirms their effectiveness degrades under random selection. This reliance on pre-defined operators limits the approach's generalizability and ability to discover novel, unexpected collaboration strategies outside of a few common plan-editing patterns. Do the authors have experiments/scenarios where they saw this behaviour?
- The Memory-guided Evaluation relies on retrieving transitions with similar object locations and interaction failures to judge the viability of a counterfactual plan. This suggests the framework's effectiveness may be sensitive to memory density and the quality of the embedding function ($e(\cdot)$ in Eq. 1) used for retrieval, neither of which is thoroughly analyzed.
- Scaling beyond three agents introduces diminishing returns, with performance slightly decreasing when moving from three to four agents. This suggests that the current constraint set or the evolutionary search complexity may struggle to manage the increased coordination complexity and load imbalance of larger teams.

**Questions:**

1. Since the Symbolic Constraint Inductor is a critical component, could the authors provide a quantitative measure of its fidelity? Specifically, how often does the LLM-based $\mathcal{C}_{gen}$ module successfully induce a rule that correctly prevents a recurrence of the specific failure type, and how often does it induce a rule that is structurally irrelevant or leads to a future constraint violation?
2. The conclusion states future work involves extending counterfactual reasoning to the subtask level (e.g., goal reordering, subgoal decomposition). Given the success of the current joint action-level mutations, did the authors test a minimal experiment where they only allow mutations over the temporal ordering of the subgoals (ignoring agent assignment) to gauge the immediate benefit of this richer counterfactual space?
3. The evaluator's design is invariant under monotonic transformation, implying only the ranking is relevant. However, the ranking is driven by retrieving relevant traces and using LLM reasoning to integrate rules and outcomes. Could the authors provide a case study where the LLM's Constraint-guided Evaluation successfully rejects a candidate plan that is structurally valid but known to lead to long-term failure based on a low-support rule (i.e., a rule with low confidence/support count $\mu_r$)?
4. Given the scaling challenge observed beyond three agents, have the authors investigated whether the performance drop at four agents is correlated with an increase in Load Balance (B) violations or an increase in the number of spatial feasibility constraints being retrieved during planning, suggesting a memory/coordination constraint saturation? Are there any mitigative measures for this issue?

---

> ### Author Response · Authors · 2025-11-21
>
> We sincerely thank you for your careful review and thoughtful understanding of our work. Below, we provide detailed responses to all your comments and describe the corresponding revisions made. (Please note that all line numbers mentioned correspond to the revised PDF)
>
> ---
>
> **Weakness 1 & Question 1:**
> >*The Symbolic Constraint Induction process is not fully detailed as a black box.*
>
> >*Since the Symbolic Constraint Inductor is a critical component, could the authors provide a quantitative measure of its fidelity?*
>
> **Response**:
>
> We thank the reviewer for this important point.
>
> **1. Reproducibility of the rule induction process**: we would like to clarify that our rule induction is fully prompt-based and non-learned. Given a failure-annotated transition, the LLM outputs symbolic constraints in a fixed format. All prompts and schemas are in **`Appendix A (Previously included, see line708-730)`**. And **`Appendix B.3 (Previously included, see Table 9, lines870-905)`** further shows representative rules induced across tasks.
>
> **2. Fidelity of the Constraint Inductor**: our constraint inductor is designed to robustly guide the LLM by providing interpretable, causal rules that link failure patterns to symbolic preconditions or coordination dependencies. This structured format enables consistent rule induction across diverse scenarios and helps prevent repeated failures.
>
> **(1)** While a direct quantitative measure of “correctness” is infeasible, since rules do not directly intervene in execution, they guide counterfactual generation and evaluation. Their effect is indirect, reflected in improved plan quality. We provide qualitative examples in **`Appendix B.3 (Previously included, lines870-905)`** to show that induced rules accurately capture relevant causal dependencies across six categories (e.g., spatial feasibility, temporal consistency, load balance), confirming the inductor’s fidelity in generating meaningful constraints.
>
> **(2)** Furthermore, we include a case study in **`Figure 3 (Previously included, lines486-495)`** where a rule induced from a failure (e.g., misalignment during object pickup) leads to a constraint `OpenObject(?object) → Facing(?object)`. This constraint prevents recurrence by filtering out future plans lacking alignment, demonstrating direct impact on failure avoidance.
>
> **(3)** We also added one ablation result in **`Table 5 (New content highlighted, line436)`** , where ablating the constraint inductor `(w/o Rule Induction)`, leads to less targeted counterfactual plan generation and weaker plan selection. Results in degraded success rate (0.84 → 0.75). This highlights that symbolic constraints are essential not just for diversity, but for generating corrective and reliable plans.
>
> ---
>
> **Weakness 2:**
>
> >*The reliance on pre-defined operators limits the approach's generalizability and ability to discover novel, unexpected collaboration strategies outside of a few common plan-editing patterns. Do the authors have experiments/scenarios where they saw this behaviour?*
>
> **Response**:
>
> We propose four mutation operators, **`{SwapAgent, InsertAction, ReplaceObject, DeleteAction}`**, which are sufficient to cover all symbolic variations of plans at the action-level in the MAP-THOR domain.
>
> In MAP-THOR, each action is structured as **`Act(agent, object) `**, where the action space is discrete and verb-object grounded. Within this representation, our operator set allows the LLM to explore and recombine plan elements flexibly, without limiting the ability to generate **novel or unexpected coordination strategies.**
>
>
> We further support this with usage statistics in **`Table 2 (New Line 378–383)`**, which shows the occurrence and adoption percentage of counterfactual plans based on each operator. Notably, InsertAction accounts for 39% of mutations and 17% of selected plans, enabling nuanced refinements like spatial adjustment. The consistent contribution of all operators indicates their utility across diverse plan-editing scenarios.
>
> | Table2    | Swap | Insert |Delete| Replace|
> | -  |-  | -  |- | - |
> | Occurrence Frequency  | 12% | 39% | 22%  | 27%  |
> | Adoption Frequency |  8%  |   17%   |     8%      |   10%    |
>
> Finally, we respectfully note that our focus is not on discovering new mutation operators, but on using symbolic constraints to guide reliable plan refinements. As also observed in prior work on evolutionary planning (e.g., EvoFlow [1]), most frameworks operate with fixed operator sets and focus search via evaluation. That said, we agree that discovering or learning new operators at the subgoal or program level is a promising direction, which we plan to explore in future work.
>
> ---

---

> > ### Author Response · Authors · 2025-11-21
> >
> > **Question 2**
> >
> > >*The conclusion states future work involves extending counterfactual reasoning to the subtask level (e.g., goal reordering, subgoal decomposition). Given the success of the current joint action-level mutations, did the authors test a minimal experiment where they only allow mutations over the temporal ordering of the subgoals (ignoring agent assignment) to gauge the immediate benefit of this richer counterfactual space?*
> >
> > **Response**:
> >
> > Thank you for this insightful eequestion, we truly appreciate your suggestion to explore counterfactual mutations at the subgoal level. This is a fascinating direction that opens up new dimensions for structured plan optimization.
> >
> >
> > Our choice to operate at the **action level** stems from a core motivation: in **partially observable**, **physically grounded** environments like MAP-THOR, reasoning at the action granularity helps minimize the **planning-to-execution gap**. Fine-grained edits (e.g., adjusting movement or manipulation actions) allow agents to recover from failure while preserving physical feasibility and inter-agent coordination. This level of reasoning is most effective for handling real-time constraints, spatial conflicts, and low-level execution errors common in embodied multi-agent settings.
> >
> > That said, we fully agree that evolution over subgoal structures, such as reordering, decomposing, or substituting subgoals, offers a richer and complementary search space. However, our current framework does not explicitly model subgoal-level structure, plans are represented as flat sequences of agent-grounded actions without hierarchical decomposition. we plan to actively explore it in future work.
> >
> > ---
> >
> > **Question 3:**
> > >*Case study: Rejection of valid plans based on low-support constraints.*
> >
> > **Response**:
> > We thank the reviewer for the thoughtful question. **`Figure 3 (line486-495)`** illustrates this behavior in the `“Turn on all stove knobs” task.` In Step 5, the original plan `(NavigateTo(stove_2), Open(stove_1))` fails due to **collision** and **misalignment**. From this single failure, EvoCF induces two low-support constraints:
> > >`NavigateTo(?Object) → Avoid(CollisionWith(?Agent))`
> > `OpenObject(?Object) → Facing(?Object).`
> >
> > In Step 6, a structurally valid plan `(MoveAhead, Open(stove_1))` is generated but rejected by the evaluator in favor of CF Plan A `(Rotate(30°), MoveLeft)`, which avoids the earlier issues. This leads to successful execution, showing how even a low-support rule can help reject plausible but failure-prone plans.
> >
> > ---
> >
> > **Question 4:**
> > >*Have the authors investigated whether the performance drop at four agents is correlated with an increase in Load Balance (B) violations or an increase in the number of spatial feasibility constraints being retrieved during planning. Are there any mitigative measures for this issue?*
> >
> >
> > **Response**:
> >
> > Thank you for this thoughtful observation. In line with your intuition, we did observe that the drop in success rate at 4 agents correlates with a decline in **Load Balance** (B: 0.81 → 0.74). This reflects increased spatial contention: with four agents operating in cluttered, partially observable environments, the number of spatial feasibility constraints (e.g., blocked paths, misalignment) rises. As a result, agents require more steps to coordinate access and avoid collisions, leading to longer, but still executable plans. Since MAP-THOR enforces a hard cap of **L=30** steps per episode, most of the failures at 4 agents are due to exceeding this cap, not plan infeasibility.
> >
> > One potential mitigation is to assign tasks based on spatial proximity as the number of agents increases. In crowded environments, proximity-aware task allocation can reduce congestion and redundant movement, while also avoiding the generation of idle plans. This not only improves spatial efficiency but also helps maintain better load balance.
> >
> > ---
> > We are deeply grateful for Reviewer pAHc’s thoughtful and intellectually generous review. Your insights meaningfully strengthened our understanding of the problem space, we would be glad to continue the conversation on any remaining points you find important.
> >
> > ---
> >
> > [1] Zhang G, Chen K, Wan G, et al. Evoflow: Evolving diverse agentic workflows on the fly[J]. arXiv preprint arXiv:2502.07373, 2025.

---

### Official Review · Reviewer_Zhcg · 2025-11-04

**Soundness:** 3
**Presentation:** 3
**Contribution:** 2
**Rating:** 6
**Confidence:** 3

**Summary:**

The paper introduces EvoCF, a hybrid framework to improve multi-agent planning for embodied AI. It addresses the common failure of Large Language Model (LLM) planners to account for real-world physical and coordination constraints. EvoCF works in three stages: (1) it learns a library of symbolic rules by observing past failures; (2) it uses an evolutionary generator, guided by these rules, to create many "what-if" (counterfactual) plan variants; and (3) it uses an experience-driven evaluator to select the most robust plan.

**Strengths:**

- It tackles the "short-sightedness" of LLM planners, which often fail to maintain context across multi-step tasks.

- The hybrid approach of learning symbolic rules from failure and using them to guide an evolutionary search is a new and effective way to explore the planning space.

**Weaknesses:**

- The paper's "symbolic rules" (Table 5) appear to be simple, correlational heuristics learned post-hoc from failures (e.g., 'being near X and Y causes interference'). This seems fundamentally different from a true causal understanding of why the failure occurred (e.g., 'Agent 1's path requires the same 3D space Agent 2 currently occupies'). Is the system simply learning brittle heuristics that only prevent seen failures, or can it prove it generalizes to novel failure modes? How would EvoCF handle an entirely new type of constraint (e.g., a "social" rule like "don't enter the room while the user is in it") that it has never seen fail before?

- The 'evolutionary generator' uses simple mutation operators (Swap, Insert, Delete) to explore the plan space. This type of local search is prone to getting stuck in local optima; it might find a 'less bad' plan that avoids a known rule but could miss a globally optimal plan (e.g., one that is dramatically faster). Did the authors investigate whether EvoCF produces plans that are merely sufficient versus those that are provably optimal? How does this method compare to more traditional, exhaustive planners that can guarantee optimality?

- The paper presents EvoCF as a hybrid LLM planner. However, the LLM's initial plan is explicitly 'distrusted' and immediately "mutated" by a symbolic, rule-based system. The "intelligence" of the solution seems to come entirely from the symbolic EvoCF component, which is a good thing. But, fundamentally, what value does the LLM provide, other than as a seed generator? Could the LLM be replaced with a simple, non-AI baseline planner (or even a hand-crafted template) with no loss in performance? What justifies this as an 'LLM-based' system?

**Questions:**

See the weaknesses. But other minor questions:

- Could the authors provide data on the planning time required by EvoCF compared to the baselines? Is there a trade-off where EvoCF takes significantly longer to find a plan, even if that plan is more efficient to execute?

- The experiments are limited to 2-agent scenarios. The evolutionary search for plans could become computationally infeasible with more agents. The authors should include some analysis of how the computational feasibility scales with the number of agents.

---

> ### Author Response · Authors · 2025-11-21
>
> We would like to express our sincere respect for your insightful review! Below, we provide detailed responses to all your comments and describe the corresponding revisions made. The newly added revisions have been updated in the PDF and highlighted for your reference.  (Please note that all line numbers mentioned correspond to the revised PDF)
>
> ---
>
> **Weakness 1:**
> >*1.1 Do the induced symbolic rules reflect true causal reasoning, or are they merely shallow, correlational heuristics learned post-hoc from failures?*
>
> >*1.2 Is the system simply learning brittle heuristics that only prevent seen failures, or can it prove it generalizes to novel failure modes?*
>
> >*1.3 How would EvoCF handle an entirely new type of constraint that it has never seen fail before?*
>
>
>
>
>
> **Response**:
>
> **1.1** While EvoCF induces symbolic rules from past failures, they are not shallow post-hoc correlations. Rule induction leverages structured memory, combining observations, actions, and failure signals, to infer why a failure occurred, e.g., via violated preconditions, spatial conflicts, or temporal misalignment. The induced rules are automatically abstracted into general symbolic forms with associated reasoning traces that explain failure mechanisms and reflect causal structure. As shown in **`Appendix B.3 (Table 9, line 870-905)`**, they capture execution-level constraints such as action feasibility, temporal dependencies, and multi-agent coordination, well beyond surface-level patterns.
>
> **1.2** EvoCF’s symbolic rules are not tied to specific tasks or instances, they are designed to be structurally reusable. Rules are induced in a typed, symbolic form (e.g., `OpenObject → Facing`) and aligned with general execution or coordination conditions . As shown in  **`Appendix B.1 (Previously incluede; Table 2, see Lines826-835)`**，rules induced from a single task significantly improve performance on structurally different, unseen tasks (+20% SR), demonstrating that the constraints capture generalizable failure patterns rather than brittle, task-specific heuristics.
>
> | Target Task | Setting | SR↑ | TR↑ |C↑| B↑| L↓ |
> | -  | -  | -  |- | - | - | - |
> | Put the pots and pans on the stove |  Planner Only  |   0.60   |     0.88      |0.95  | 0.80    |  20.13  |
> | Put the pots and pans on the stove |  w/ Transferred Constraints |   0.80   |     0.93      |   0.97    |  0.82  | 18.55 |
>
> | Target Task | Setting | SR↑ | TR↑ |C↑| B↑| L↓ |
> | -  | -  | -  |- | - | - | - |
> |Open all drawers|  Planner Only  |   0.40   |     0.75      |0.94  | 0.75   |  22.41  |
> | Open all drawers |  w/ Transferred Constraints |   0.60   |     0.88      |   0.94    |  0.79  | 20.95 |
>
>
> **1.3** EvoCF’s rule library is not fixed, it supports online updates. When a novel failure occurs, the system records the transition and induces new constraints based on the failure’s context. These are immediately integrated into the rule set and applied in subsequent planning, without retraining. Furthermore, many rule templates (e.g., mutual exclusion, temporal blocking) are structurally general and can extend to new domains. This enables EvoCF to adapt to unseen constraint types after minimal exposure, and in some cases anticipate them based on analogous structural patterns.
>
> ---

---

> > ### Author Response · Authors · 2025-11-21
> >
> > **Weakness 2:**
> >
> > >*2.1 The 'evolutionary generator' uses simple mutation operators to explore the plan space, which is prone to getting stuck in local optima. Did the authors investigate whether EvoCF produces plans that are merely sufficient versus those that are provably optimal?*
> >
> > >*2.2 How does this method compare to more traditional, exhaustive planners that can guarantee optimality?*
> >
> > **Response**:
> >
> > **2.1 “Local optima” and “merely sufficient vs. optimal”**
> >
> > **Scope and goal.** EvoCF does not claim global optimality. In multi-agent, partially observable settings with **multi-objective criteria** (e.g., feasibility, load balance, synchronization), provable optimality is ill-posed without a full world model and scalarized objective. EvoCF instead conducts a guided search over counterfactual variants and selects the top-ranked plan via an experience-driven, rule-constrained evaluator (**`Eq. 4`, line315**), which is explicitly **“world-model-free.”**
> >
> > **Why we’re not “just sufficient.”** The CF generator is not blind hill-climbing: it uses **rule** and **memory-guided** mutations that apply structural edits, allowing exploration beyond local tweaks. This enables EvoCF to escape weak configurations. The evaluator ranks candidates via symbolic constraints and retrieved outcomes, not heuristics, and consistently yields higher-quality plans. As shown in **`Table 1 (line366)`**, EvoCF improves SR by +18% over LLaMAR. Ablations in **`Table 5 (lines432-440)`** confirm both generator and evaluator contribute significantly, indicating EvoCF does not settle for merely sufficient plans.
> >
> > **2.2 Comparison to exhaustive/optimal planners**
> >
> > **Modeling assumption gap.** Exhaustive planners rely on **accurate dynamics, full observability, and scalar objectives**, assumptions incompatible with our multi-agent **POMDP** setting. EvoCF targets this regime by adopting retrieval-grounded evaluation over forward simulation, trading formal guarantees for robustness under uncertainty.
> >
> > **Computational tractability.** The joint plan space grows exponentially with agent count, making exhaustive search impractical at realistic horizons. EvoCF confines search to a rule-consistent neighborhood of the seed plan and prunes with experience-based evaluation. This provides a practical, anytime approach that scales effectively to 2–4 agents with stable performance (**`Table 3, lines393-400`**).
> >
> > **Quality–efficiency trade-off in practice.** While exhaustive planners offer optimality under strict assumptions, EvoCF demonstrates higher empirical success in MAP-THOR, with fewer planning steps than strong LLM baselines and consistent coordination across team sizes. Thus, while not provably optimal, EvoCF consistently achieves better balance, reliability, and efficiency in real-world constraints.
> >
> > **Takeaway: EvoCF is built for partially observable, multi-objective, multi-agent planning where optimality is impractical. By combining symbolic constraints and retrieval-guided evaluation, it finds stronger plans than reactive LLMs while avoiding the intractability and modeling overhead of exhaustive planners.**
> >
> > ---

---

> > > ### Author Response · Authors · 2025-11-21
> > >
> > > **Weakness 3:**
> > > >*Clarification of the LLM’s role and necessity within the hybrid planning system.*
> > >
> > > **Response**:
> > >
> > > **EvoCF is not designed to distrust or override the LLM’s plan, but to augment it through evolutionary counterfactual planning, enhancing robustness and feasibility.** This augmentation is not optional, it is essential for LLM-based planning in embodied multi-agent settings, where partial observability, spatial and temporal dependencies, and exponential joint action spaces routinely cause LLMs to fail without explicit grounding or verification. Multi-agent collaboration, in particular, is especially brittle, as failures often emerge from inter-agent conflicts that are not locally detectable at planning time.
> > >
> > > The LLM plays several core roles in EvoCF:
> > >
> > > **1.Semantic grounding:** translating natural language instructions into high-level task decompositions.
> > >
> > > **2.Agent-role assignment:** proposing structured joint plans with temporally and semantically coherent action sequences.
> > >
> > > **3. Language-conditioned generalization:** adapting to novel scenarios without hardcoded rules or supervision.
> > >
> > > These capabilities are not replaceable by template-based or symbolic planners, which lack the abstraction capacity to generalize across environments and task structures. EvoCF preserves these LLM-generated semantics and improves upon them with symbolic and experiential reasoning.
> > >
> > > In short, EvoCF demonstrates how to build on LLM generalization with structure-aware corrections, a direction that is crucial for scaling LLM-based embodied agents beyond brittle one-shot planning.
> > >
> > > ---
> > >
> > > **Question 1:**
> > > >*Could the authors provide data on the planning time required by EvoCF compared to the baselines? Is there a trade-off where EvoCF takes significantly longer to find a plan, even if that plan is more efficient to execute?*
> > >
> > >
> > > **Response**:
> > >
> > > We added per-module latency and token cost in  **`AppendixB.2 (New content highlighted; see Table7, Lines839-860)`**. Given that EvoCF employs LLaMAR as its LLM planner backbone, its additional counterfactual planning modules (rule induction, plan generation, and evaluation) introduce only a reasonable overhead in both latency and token usage. As shown in the Table below, EvoCF reduces planning steps per episode (21.87 → 18.69), helping amortize the cost across fewer, higher-quality decisions.
> > >
> > >
> > > | Table7 (seconds)   | LLaMAR-Moudles| Rule Inductor |Counterfactual Plan Generator | Evaluator| Step | Toal |
> > > | - | - | - |-| - | - | - |
> > > | LLaMAR             | 7.61 | - | -  | -  | 21.87 | 166.43 |
> > > | EvoCF(ours) | 7.61  |  1.46   |    2.17    |   2.31      |  18.69  |   269.76|
> > >
> > > **Beyond efficiency**,  this additional planning cost enables EvoCF provides two practical advantages crucial in real-world deployments: **1. Coordination stability:** higher balance and tighter synchronization reduce congestion and communication rounds, avoiding deadlocks/rollbacks. **2. Fewer within-round failures:** the evaluator prunes failure-prone plans before execution, cutting retries and safety interventions.
> > >
> > > These gains make EvoCF not only more robust, but also more cost-effective when execution and recovery are expensive.
> > >
> > > ---
> > > **Question 2:**
> > > >*The experiments are limited to 2-agent scenarios. The authors should include some analysis of how the computational feasibility scales with the number of agents.*
> > >
> > >
> > > **Response**:
> > >
> > > We would like to clarify that this experiment has already been reported **`(Previously included, Table 3 (Line393-400)`**. As shown in Table 3, EvoCF performs consistently across 2, 3, and 4-agents settings: SR improves to 0.87 with 3 agents and remains high (0.82) with 4. The drop in balance and slight rise in planning steps at 4 agents reflect coordination complexity under the fixed step cap (L=30), not computational limits.
> > >
> > > | # Agents| SR↑ | TR↑ |C↑| B↑| L↓ |
> > > | -  | -  | -  |- | - | - |
> > > | 2 |  0.84  |   0.95   |     0.99      |   0.89    |  18.69  |
> > > | 3 |  0.87  |   0.96   |     0.99      |   0.81    |  17.36  |
> > > | 4 |  0.82  |   0.93   |     0.99      |   0.74    |  19.51  |
> > >
> > > ---
> > > We thank Reviewer Zhcg for your encouraging and perceptive review. Your feedback prompted valuable improvements in both our empirical evaluation and theoretical framing. We would be glad to continue the conversation on any remaining points you find important.

---

### Meta-Review · Area_Chair_En8k · 2026-01-07

**Summary:**

The paper proposes EvoCF, a framework designed to improve multi-agent embodied planning. The method integrates three main modules: a Symbolic Constraint Inductor (which learns rules from failure), an Evolutionary Counterfactual Plan Generator (which uses mutation operators to explore plan variants), and a Retrieval-Augmented Evaluator. The authors validate the approach on the MAP-THOR benchmark, demonstrating performance gains over the LLaMAR baseline.

The decision to recommend rejection is based on the following key concerns shared by the reviewers:
1.  **Limited Technical Novelty and Heuristic Nature:** Multiple reviewers (Zhcg, 1hYG, pAHc) pointed out that the core components rely heavily on hand-crafted heuristics. Specifically, the "evolutionary" generator uses a small set of predefined mutation operators (Swap, Insert, Delete), and the "symbolic rules" act more as shallow correlational templates than deep causal reasoning. While effective on the specific benchmark, this design is viewed as an engineering integration of known techniques rather than a fundamental algorithmic advancement suitable for ICLR.
2.  **Narrow Evaluation Scope:** The experimental validation is restricted to a single simulation benchmark (MAP-THOR) with a limited number of agents (2-4). Reviewers (HnHQ, 1hYG) expressed concerns about scalability and generalization. The rebuttal did not demonstrate the framework's effectiveness across diverse domains or larger-scale systems, leaving the robustness of the "general framework" claim unsupported.

**Reviewer Concerns:**

### Addressed Concerns:
-  **Missing Citations:** The authors successfully integrated the missing MACI references and clarified the differences between their embodied focus and the existing work's focus on dialogue/reasoning.
-  **Computational Cost & Ablations:** The authors provided the requested data on token usage, latency, and module-level ablations, which clarified the system's efficiency.
-  **Clarifications:** Notation issues (raised by HnHQ) and specific implementation details were clarified in the revision.

### Outstanding Concerns:
-  **Dependence on Manual Heuristics:** The concern that the method is "brittle" or heavily reliant on the manually designed mutation operators and constraint templates remains. The rebuttal did not show how the system would discover novel strategies outside these pre-defined bounds.
-  **Scalability:** The constraint of the evaluation to small-scale scenarios (2-4 agents) remains a limitation. The argument that "the benchmark limits this" confirms the evaluation's narrow scope rather than mitigating the concern.
-  **Depth of Contribution:** The perception that this is a "focused engineering integration" (WJsZ) rather than a novel learning paradigm persists.

**Reviewer Scores:**

- **Reviewer WJsZ (Score: 2->4):** The fundamental critique regarding the work being an "uncredited reinvention" or engineering effort likely limits the score improvement to a borderline level (4).
- **Reviewer HnHQ (Score: 4->4):** Likely remains a **4**. The rebuttal clarified definitions but did not resolve the fundamental concern regarding the limited experimental scope (single dataset, few agents).
-  **Reviewer 1hYG (Score: 4->4):** Likely remains a **4**. The reviewer explicitly questioned the expressiveness of the small set of mutation operators. The rebuttal confirmed these are manually designed, which reinforces the concern about limited generalizability.
- **Reviewer Zhcg (Score: 6) & pAHc (6->6):** These scores likely remain at **6**. They appreciated the empirical results but noted the "shallow" nature of the rules. In the context of the strong critiques from others, their support is not strong enough to champion acceptance.

---

### Decision · Program_Chairs · 2026-01-26

Reject